# A clinically applicable and scalable method to regenerate T-cells from iPSCs for off-the-shelf T-cell immunotherapy

Shoichi Iriguchi[1,2,7], Yutaka Yasui[1,7], Yohei Kawai[1,2,7], Suguru Arima[2,3], Mihoko Kunitomo[2,3], Takayuki Sato[2,3], Tatsuki Ueda[1], Atsutaka Minagawa[1,2], Yuta Mishima [1,2], Nariaki Yanagawa[1,2], Yuji Baba[2,3], Yasuyuki Miyake[1,2], Kazuhide Nakayama[2,3], Maiko Takiguchi[2,3], Tokuyuki Shinohara[2,3], Tetsuya Nakatsura[4], Masaki Yasukawa [5,6], Yoshiaki Kassai[2,3], Akira Hayashi[2,3] & Shin Kaneko [1,2 ✉]

Clinical successes demonstrated by chimeric antigen receptor T-cell immunotherapy have facilitated further development of T-cell immunotherapy against wide variety of diseases. One approach is the development of "off-the-shelf" T-cell sources. Technologies to generate T-cells from pluripotent stem cells (PSCs) may offer platforms to produce "off-the-shelf" and synthetic allogeneic T-cells. However, low differentiation efficiency and poor scalability of current methods may compromise their utilities. Here we show improved differentiation efficiency of T-cells from induced PSCs (iPSCs) derived from an antigen-specific cytotoxic T-cell clone, or from T-cell receptor (TCR)-transduced iPSCs, as starting materials. We additionally describe feeder-free differentiation culture systems that span from iPSC maintenance to T-cell proliferation phases, enabling large-scale regenerated T-cell production. Moreover, simultaneous addition of SDF1α and a p38 inhibitor during T-cell differentiation enhances T-cell commitment. The regenerated T-cells show TCR-dependent functions in vitro and are capable of in vivo anti-tumor activity. This system provides a platform to generate a large number of regenerated T-cells for clinical application and investigate human T-cell differentiation and biology.

[1] Shin Kaneko Laboratory, Department of Cell Growth and Differentiation, Center for iPS Cell Research and Application (CiRA), Kyoto University, Kyoto, Japan. [2] Takeda-CiRA Joint Program (T-CiRA), Fujisawa, Japan. [3] T-CiRA Discovery, Takeda Pharmaceutical Company, Fujisawa, Japan. [4] Division of Cancer Immunotherapy, Exploratory Oncology Research & Clinical Trial Center, National Cancer Center, Kashiwa, Chiba, Japan. [5] Department of Hematology, Clinical Immunology and Infectious Diseases, Ehime University Graduate School of Medicine, Toon, Japan. [6] Ehime Prefectural University of Health Sciences, Tobe, Japan. [7] These authors contributed equally: Shoichi Iriguchi, Yutaka Yasui, Yohei Kawai. ✉email: kaneko.shin@cira.kyoto-u.ac.jp

Clinical efficacy of chimeric antigen receptor (CAR)-transduced T-cell immunotherapy has demonstrated that T-cells can be used to treat a wide variety of diseases. One approach for broadening the use of T-cell immunotherapy may be the development of "off-the-shelf" T-cell sources. To date, candidate cells for such T-cell sources include peripheral blood T-cells from healthy donors and T-cells generated form pluripotent stem cells (PSCs)[1].

T-cells derived from PSCs provide a versatile platform to study human T-cell biology and an alternative cell source for allogeneic T-cell immunotherapy. In vitro differentiation of T-cells from human embryonic stem cells (ESCs) has been demonstrated by co-culturing ESC-derived hematopoietic progenitor cells (HPCs) or hemogenic endothelial cells (HECs) with either delta-like (DL)-1 or DL-4-expressing OP9 stroma cells[2,3]. Generation and re-differentiation of induced PSCs (iPSCs) reprogrammed from antigen-specific cytotoxic T-cell clones and their anti-tumor activities in preclinical models have paved the way to clinical translation of PSC-derived T-cell immunotherapy[4–9]. Generation of T-cells from PSCs consists of a series of differentiation processes with many key intermediates. These processes are achieved by recapitulating signal transduction events during human hematopoietic development and T-cell differentiation from hematopoietic stem and progenitor cells. PSCs are first induced to differentiate into mesodermal progenitors expressing kinase insert domain receptor (KDR), followed by HECs. HECs generate HPCs through the process known as endothelial-to-hematopoietic transition[10,11]. In the presence of notch signaling mediated by DL ligands, the resulting HPCs differentiate sequentially into progenitor T-cells, immature single-positive cells, and $CD4^+CD8^+$ double-positive (DP) T-cells; these events have also been observed in human thymus[12–14]. In addition to signals mediated by direct interactions between thymic stromal cells and differentiating thymocytes, several soluble factors in the thymic niche such as IL-7, FMS-like tyrosine kinase 3 ligand (FLT3L), stem cell factor (SCF), and chemokines CXCL12 and CCL25[15] have been identified to play a pivotal role. The final stage of T-cell differentiation involves maturation of DP-cells into $CD4^+$ or $CD8^+$ single-positive (SP) T-cells. Although protocols to induce maturation of CD8 SP T-cells from iPSCs are readily available, in vitro maturation methods to induce CD4 SP T-cells have not been achieved, except for an organoid-based T-cell differentiation method[16].

Use of multiple murine stromal feeder layers during T-cell differentiation may limit the development of "off-the-shelf" T-cell sources for allogeneic T-cell immunotherapy. Standard T-cell differentiation protocols require different types of murine feeders in each stage of differentiation including mouse embryonic feeder (MEF) for PSC proliferation, OP9 or C3H10T1/2 feeder for hematopoietic progenitor induction, and OP9-DL1 feeder cells for T-cell differentiation. Each feeder requires different sets of serum and basal media for maintenance culture and co-culture with differentiating PSCs, making the control and reproducibility challenging. Moreover, T-cells derived from iPSCs are proliferated on human peripheral blood mononuclear cell (PBMC) feeder cells or immortalized cell lines expressing receptor-specific antigens (i.e., K562 cell lines exogenously expressing CD19). For safety measures, all these cells would need to be free of virus contamination for clinical application, which may be highly expensive or impossible in some cases. Thus, development of feeder-free (Ff) differentiation system during all stages of differentiation is critical to develop iPSC-based T-cell immunotherapy. Although feeder- and serum-free cultures for PSC proliferation and hematopoietic differentiation are well-established, no such culture systems for T-cell differentiation and proliferation are available.

In this study, we report the development of a highly efficient and scalable method for T-cell generation from human iPSCs. This system allows us to generate T-cells from iPSCs derived from T-cell clones and T-cell receptor (TCR)-engineered iPSCs in the absence of feeder layers in all stages of differentiation. Using this system, we identified SDF1α and a p38 inhibitor to be the factors that facilitate T-cell differentiation. TCR-engineered iPSC-derived T-cells using the system show anti-tumor activities both in vitro and in vivo tumor xenograft model. This system may provide a foundation to translate allogeneic iPSC-derived T-cell immunotherapy to the clinic as well as to study human T-cell biology.

## Results

**Generation of hematopoietic progenitors with lymphoid potential from iPSCs by modified embryo body formation.** We aimed to develop an efficient and scalable Ff system that produces cytotoxic T-cells from iPSCs. To begin with, we divided the whole process into four steps and optimized each step, which include induction of HPCs with lymphoid potentials from iPSCs, differentiation of HPCs into $CD4^+CD8αβ^+$ double-positive (DP) cells, maturation of DP-cells into $CD8αβ^+$ single-positive (SP) T-cells, and finally proliferation of CD8αβ SP T-cells (Supplementary Fig. 1). Optimizations were first carried out with an iPSC line derived from peripheral blood T-cells (TkT3V1-7)[4]. For hematopoietic induction, we choose the formation of well-established embryoid bodies (EB) with some modifications as depicted in Fig. 1a[3,17,18]. To initiate differentiation, iPSCs maintained on the Laminin-511-based culture system[19] were dissociated into a single-cell suspension and seeded into a well of an ultra-low attachment plate to facilitate EB formation and cultured for 24 h in a hypoxic incubator. EB formation and subsequent hematopoietic differentiation of single-cell dissociated PSC were greatly improved by using iPSC proliferation medium and addition of CHIR99021. Differentiating EBs were analyzed at defined time points for the expression of CD34 and CD43 by flow cytometry (Fig. 1b). As expected, differentiating EBs upregulated the expression of CD34 for the first 4 days after differentiation. $CD43^+$ cells, which marks emerging hematopoietic cells[20], appeared on day 6 and this population steadily increased over the next week. By day 14 after differentiation, the total cell numbers in culture reached to $5 × 10^6$ cells with slight variations by iPSC lines (Supplementary Fig. 2a). Similar results were also obtained by using an embryonic stem cell line (KhES-3).

The progenitor activity of differentiating EBs were evaluated by colony-forming-unit (CFU) assay by FACS sorting. Colony-forming and myeloid multilineage potentials were exclusively limited to $CD34^{med}CD43^+$ fraction, suggesting that iPSC-HPCs (iHPCs) were enriched in this fraction (Supplementary Fig. 2b). Finally, we examined whether iHPCs could differentiate into the T-cell lineage under Ff culture conditions using immobilized-delta like 4 (DL4) protein as previously described using cord-blood hematopoietic stem and progenitor cells (CB-HSPCs)[21,22]. Notch signaling transduced by DL1 and DL4 is indispensable to human thymocyte development and its absence in lymphoid progenitors results in their development to B-cells[13]. Moreover, we also used retronectin, a fragment of fibronectin, together with DL4 because it has been shown to improve T-cell differentiation from CB-HSPCs[22]. After 21 days of differentiation with medium changes and weekly transfer of the differentiating cells as shown in Fig. 1c, iHPCs were able to give rise to $CD7^+CD5^+$ T-cell progenitors as well as more mature $CD4^+CD8αβ^+$ DP-cells under the Ff culture condition (Fig. 1d, e). Fractionation of differentiating EBs for T-cell potential assessment revealed that only the $CD34^{med}CD43^+$ fraction contained T-cell potential

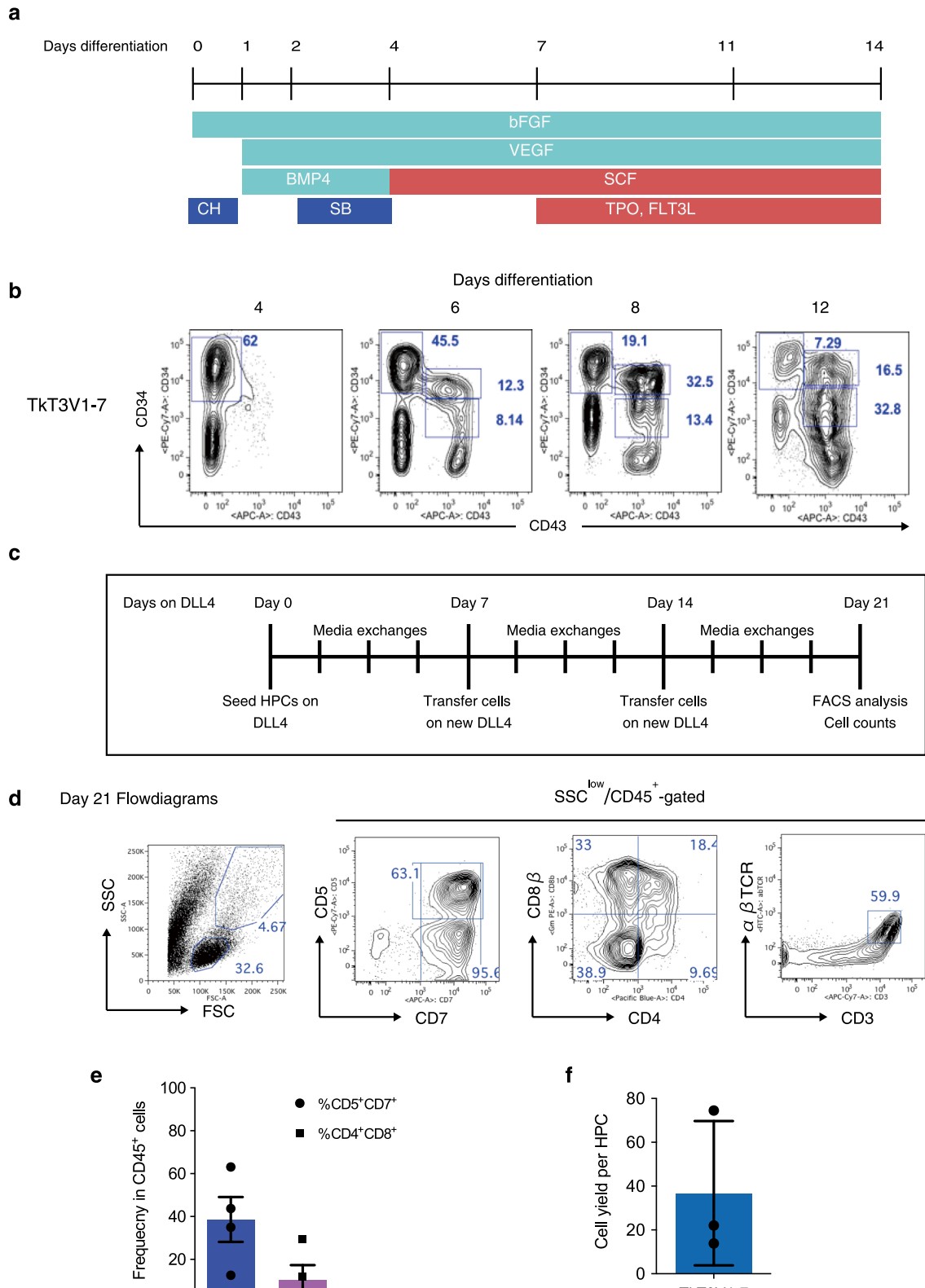

under the Ff condition, consistent with the CFU assay results (Supplementary Fig. 2c, d). However, compared to the efficiency and yield of hematopoietic differentiation, efficiency and yield of T-cell differentiation were low with high variability by differentiation lots (Fig. 1f). These results demonstrate the feasibility of progenitor T-cell generation from iPSCs in a feeder-cell-free condition, but indicate that further refinements are necessary for T-cell differentiation to achieve relevant number of cells for clinical translation.

**Optimization of T-cell differentiation cultures for iHPCs**. To increase the robustness, reproducibility, and yield of T-cell

**Fig. 1 Generation of CD4$^+$CD8$\alpha\beta^+$ DP T-cells from T-iPSCs in feeder-free (Ff) culture conditions. a** Scheme of hematopoietic induction from T-iPSC clones. EBs were generated from a single-cell dissociated human T-iPSCs proliferated in a feeder- and serum-free condition and induced to differentiate into mesoderm and later into hematopoietic cells in the presence of CH, BMP-4, bFGF, and VEGF. The emerging hematopoietic cells were proliferated in the presence of hematopoietic cytokines. SB was added on day 2 to induce definitive hematopoiesis. CH, CHIR99021; SB, SB431542. **b** Representative kinetic analysis of hematopoietic differentiation from a T-iPSC clone, TkT3V1-7, assessed based on the expression levels of CD34 and CD43 at the indicated days after induction ($n = 3$). **c** Schedule of T-cell differentiation of T-iPSC-HPCs in immobilized-DL4 and retronectin culture plates. **d** Representative flow cytometry plots of T-iPSC-HPCs differentiated on DL4 and RN cultures in the presence of SCF, TPO, FLT3L, and IL-7 for 21 days ($n = 4$). **e** Frequencies of CD5$^+$CD7$^+$ progenitor T-cells (blue) and CD4$^+$CD8$\alpha\beta^+$ DP-cells (purple) generated from T-iPSC-HPCs 21 days after differentiation on DL4 and RN cultures ($n = 4$). **f** DL4 and RN cell yields after 21 days of culture normalized to the input T-iPSC-HPC numbers ($n = 3$). Data represent mean ± SD of $n$ independent experiments.

production from iHPCs derived from several antigen-specific T-cell clones, we tested additional factors that are known to play pivotal role in thymocyte development. Of all the tested factors using the TkT3V1-7 line, the combination of CXCL12, also known as SDF1α, and a p38 inhibitor, SB203580, in synergy showed greatest effect on enhancing cell yields and frequencies of DP and immature single positive (ISP) cells, while inhibiting frequencies of CD4$^-$CD8$^-$ double-negative (DN) and CD8SP cells. Furthermore, addition of both reagents inhibited apoptosis of differentiating cells as they differentiated from DN to DP cells. In order to evaluate the effect of each reagent, we performed flowcytometric analysis of differentiating cells for the expressions of T-cell differentiation markers, CD4, CD5, CD7, and CD8β and apoptosis-related markers, Annexin V and 7-AAD at different time points during differentiation. As shown in Fig. 2a, differentiating cells first acquired CD7 expression during the first two weeks and subsequently upregulated CD5, and finally gained CD4 and CD8b expressions, indicating this in vitro differentiation culture recapitulates dynamics of thymocyte development in terms of surface marker expression transitions (Fig. 2a). Analysis of apoptosis markers indicated that addition of both agents and SB203580 alone inhibited apoptosis of differentiating cells from the 2$^{nd}$ week to the 3$^{rd}$ week of culture (Fig. 2b). Addition of both reagents and SB203580 alone also enhanced acquisition of CD7 expressions as early as one week after differentiation (Fig. 2c). Moreover, frequency of DP cells was improved only when both reagents were present (Fig. 2c). Cell yields were highest when both SDF1 and SB203580 were added in the culture, indicating its synergistic effect (Fig. 2d).

We next investigated the molecular mechanism by which these factors enhance T-cell differentiation. Measurement of gene expression involved in T-cell differentiation (*BCL11B*, *GATA3*, *TCF7*, *TCF3*, *NOTCH1*, *HES1*, and *DTX1*) and myeloid commitment (*SPI1*) of sorted iHPC in the presence or absence of the two factors for the first 48 and 96 hours revealed that *BCL11B*, the master regulator of T-cell commitment, and several NOTCH signaling targets (*GATA3*, *TCF7*) were significantly upregulated in cells exposed to the two factors by as early as 48 h after induction compared to that in cells not exposed to the two factors (Fig. 2e). On the other hand, a myeloid differentiation transcription factor, *SPI1* was significantly decreased in cells exposed to the two factors compared to that in cells not exposed to the two factors (Fig. 2e). Thus, we identified SDF1α and SB203580 as factors that enhance T-cell differentiation and cell yield from iHPCs by strongly activating T-cell commitment genes.

We also tested if the feeder-free culture can be extended to serum-free medium conditions. Among the serum-free media we tested, iHPCs could be differentiated into DP cells in αMEM supplemented with bovine serum albumin, insulin, and transferrin. In other media, iHPCs differentiated into CD7$^+$ cells, but not into DP cells (Supplementary Fig. 3).

**Generation and functional assessment of CD8 SP T-iPSC-CTLs.** To generalize the utility of the two factors and to examine antigen-specific in vitro functions of regenerated mature CD8αβ$^+$ SP T-cells, we induced three iPSC clones derived from cytotoxic T-cell (CTL) clones with known antigen-specificity (T-iPSCs) for differentiation into mature T-cells. We used two T-iPSC clones derived from CTL clones specific to a Nef38-8 peptide$_{138-145}$ and a Gag28-8 peptide (KYKLKHIVW) of human immunodeficiency virus (HIV)-1 (H25-4 and H25-31, respectively) and a CTL clone specific to a glypican-3 (GPC3) peptide$_{144-152}$ since in vitro functions of both original CTL clones and regenerated T-iPS-T-cells have been demonstrated[4,9]. We used recombinase-activating-gene (RAG)-2-deleted GPC3 T-iPSC clone to prevent additional endogenous TCR rearrangement during T-cell differentiation, which would lead to loss of antigen-specificity as T-iPSCs still contain the endogenous TCR. By day 18 after differentiation, all three differentiating T-iPSC clones acquired expression of both CD34 and CD43, consistent with the TkT3V1-7 clones (Supplementary Fig. 4a). The number of cells after 18 days of culturing was comparable among the three T-iPSC lines (Supplementary Fig. 4b). However, the frequency and yield of CD34$^+$CD43$^+$-HPCs was lower in culture with GPC3 T-iPSC than those of the other two clones at day 18 as GPC3 T-iPSCs differentiate faster than the other clones (Supplementary Fig. 4 c, d). T-iPSC-HPC successfully differentiated into CD7$^+$CD5$^+$ T-cells, a subset of which were also CD4$^+$CD8αβ$^+$ DP cells, by day 21 after T-cell differentiation in the presence of SDF1α and SB203580, demonstrating the utility of SS for other iPSC clones (Fig. 3a, b). We also induced the maturation of DL4 cells into CD4$^-$CD8αβ$^+$ mature T-cells by TCR stimulation as previously described[9] (Fig. 3c). After stimulation for 7 days, DL4 cells completely differentiated into CD4$^-$CD8αβ$^+$ SP T-cells (T-iPSC-iCD8αβ T-cells). Importantly, most of the iCD8αβ T-cells were positive to their specific tetramers (Fig. 3d). iCD8αβ T-cells at this stage expressed CD45RA and CD27, but not CCR7 and CD62L (Supplementary Fig. 5a). The yields from T-iPSC to T-iPSC iCD8αβ T-cells were estimated to be 73,275, 20,107, and 2384.7 for H25-4, H25-31, and GPC3 T-iPSCs, respectively (Fig. 3e). These results confirmed the successful generation of CD8 SP T-cells from multiple T-iPSC lines.

We next explored the proliferation potential of the three T-iPSC iCD8αβ T-cells by TCR stimulation and antigen-specific cytotoxicity. Upon the TCR stimulation, all T-iPSC iCD8αβ T-cells steadily proliferated to average 400-fold 14 days after stimulation (Supplementary Fig. 5b). Flow cytometric analysis showed that most of the proliferated cells were tetramer$^+$ CD8αβ$^+$, but lost CD27 expression (Supplementary Fig. 5c, d). Finally, we tested the antigen-specific cytotoxicity of the proliferated T-iPSC iCD8αβ T-cells. H25-4-derived T-iPSC iCD8αβ T-cells induced apoptosis of Nef peptide-loaded lymphoblastoid cell lines (LCLs), but not Gag peptide-loaded LCLs (Fig. 3f). On the contrary, H25-31-derived T-iPSC iCD8αβ T-cells killed Gag peptide-loaded LCLs, but did not show such

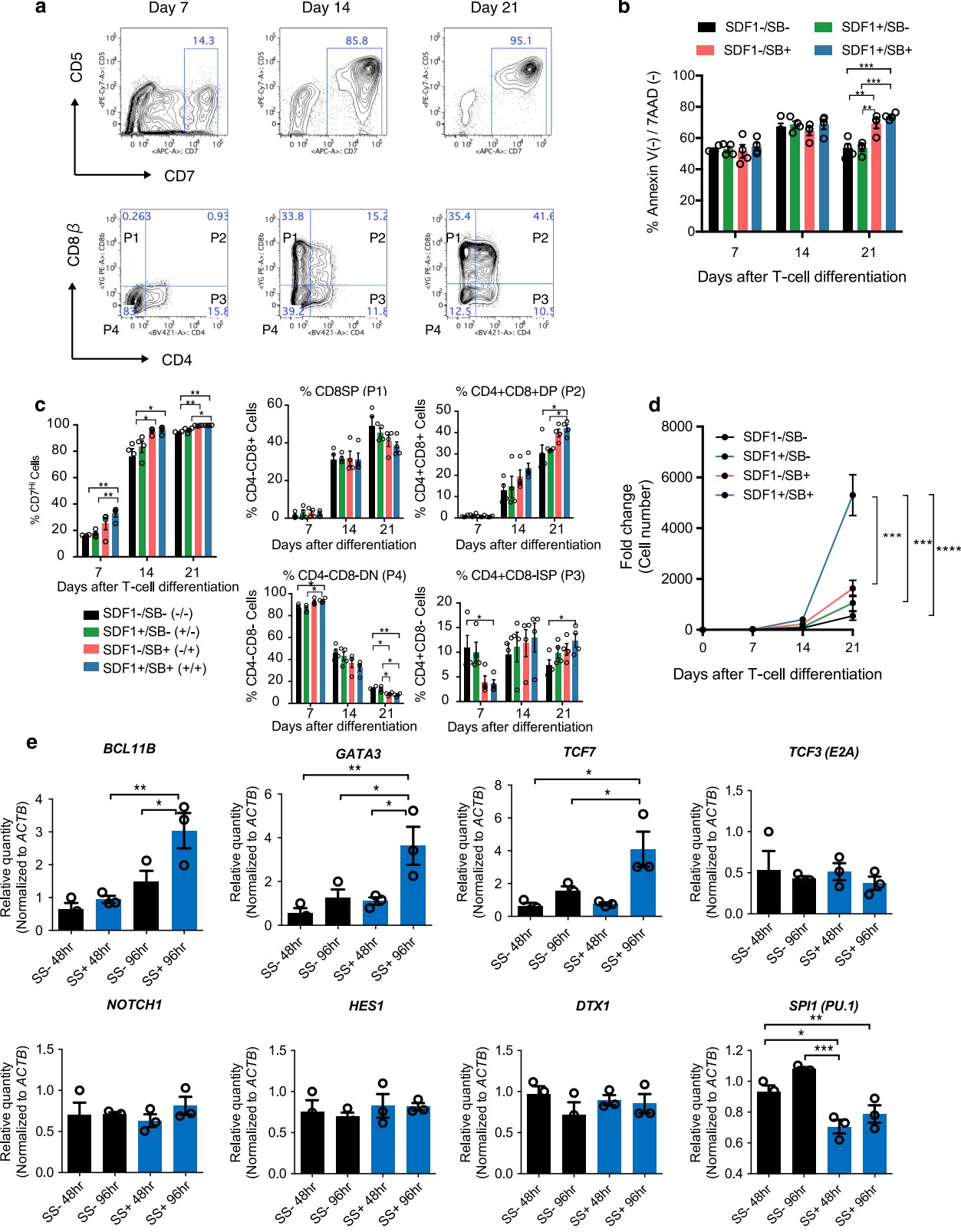

**Fig. 2 Optimization of T-cell differentiation in DL4 and RN cultures. a** Representative kinetic flow cytometric analysis of T-cell differentiation on DL4 and RN cultures assessed at the indicated days. **b** Frequencies of Annexin V(-)/7-AAD (-) live cells in differentiating cultures over 3 weeks in the presence or absence of SDF1α and/or SB203580 ($n = 4$). **c** Frequencies of the indicated cell types defined as in (**a**) in the differentiation culture over 21 days in the presence or absence of SDF1α and/or SB203580 ($n = 4$). **d** Kinetics of differentiating T-iPSC-HPC counts over 21 days on DL4 and RN in the presence or absence of SDF1α and/or SB203580 ($n = 4$). **e** Gene expression related to T-cell differentiation, Notch signaling pathway, and a myeloid lineage gene (SPI1 (PU.1)) detected by qPCR after T-iPSC-HPCs were cultured for 48 or 96 h on DL4 and RN $+/-$ SS ($n = 3$). Data represent mean ± SEM of $n$ independent experiments. *$P < 0.05$; **$P < 0.01$; ***$P < 0.001$ by one-way ANOVA with Tukey's multiple comparison test.

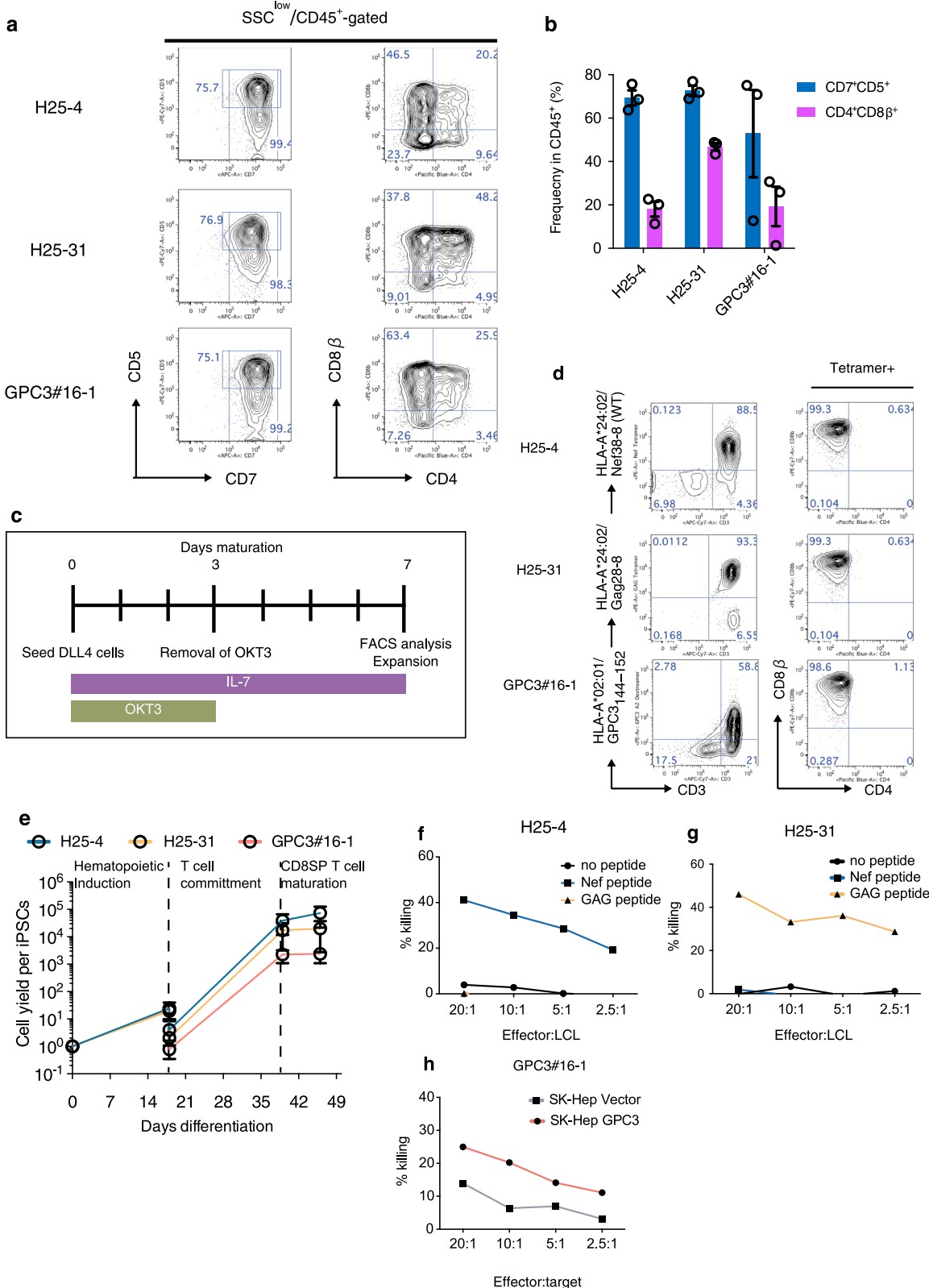

activity when co-cultured with Nef peptide-loaded LCLs (Fig. 3g). Similarly, GPC3-derived T-iPSC iCD8αβ T-cells induced apoptosis of the SK-Hep cell line exogenously expressing GPC3, but showed little activity against the SK-Hep transduced with an empty vector (Fig. 3h). Thus, we demonstrated the generation and functions of CD8αβ + T-cells derived from multiple T-iPSC clones in a Ff differentiation culture system.

**iCD8αβ T-cell generation from TCR-engineered HLA-homozygous iPSCs.** We also investigated whether the differentiation system could be extended to TCR-engineered iPSC lines. For this, we used an iPSC clone generated from a human leukocyte antigen (HLA) homozygous donor in our institute[23–25]. The scheme of production for TCR-engineered iCD8αβ T-cells are illustrated in Fig. 4a. To mention briefly, HLA-homozygous iPSCs (Ff-I01s04)

**Fig. 3 Generation of CD8αβ$^+$ T-cells from antigen-specific T-iPSCs in the optimized condition. a** Representative flow cytometry plots of three T-iPSC-derived differentiating cells after 21 days in SS + condition showing expression levels of CD5 vs CD7 and CD8αβ vs CD4. **b** Frequencies of CD5$^+$CD7$^+$ progenitor T-cells (blue) and CD4$^+$CD8αβ$^+$ DP-cells (purple) generated from the indicated T-iPSC-HPCs after 21 days on DL4/RN + SS condition ($n = 3$). **c** Schedule of maturation of DL4-cells to induce CD4$^-$CD8αβ$^+$ T-cells. **d** CD4$^-$CD8αβ$^+$ T-cell yield as determined by input T-iPSC count during differentiation processes. **e** Representative flow cytometry plots obtained 42 days after differentiation (after 7 days in maturation culture as shown in (**c**)) showing the expression levels of tetramer and CD3 (left) and CD8β and CD4 (right) of regenerated T-cells ($n = 3$). Data represent mean ± SEM of $n$ independent experiments. **f–h** In vitro cytotoxicity assay of proliferated CD8αβ$^+$ SP T-cells measured by $^{51}$Cr release assay using LCL cells loaded with specific peptides (Nef38 for H25-4 (**f**) and Gag28-8 for H25-31 (**g**)) or SK-Hep cells transduced with GPC3 or empty vectors (GPC3#16-1 (**h**)) as target cells. Data are representative of two independent experiments.

were transduced with a lentivirus vector harboring α and β chains of a HLA-A*24:02-restricted TCR specific to the Wilms tumor 1 (WT1)$_{235–243}$ peptide (WT1-FfI01s04) [9,26,27] and transgene-positive cells were isolated by FACS. After colony selection, we choose #4-2 clone for further investigation. The 4-2 clone successfully differentiated into HPC comparable to the untransduced parental HLA-homo iPSC line, FfI01s04 (Fig. 4b). Furthermore, iHPCs derived from the 4-2 clones could differentiate into CD4$^+$CD8αβ$^+$TCRαβ$^+$ T-cells by 21 days in the DL4 + SS condition, whereas those differentiated from the parental iHPCs generated TCRγδ$^+$ T-cells in the same culture condition, indicating that TCRαβ-engineering facilitated TCRαβ T-cell generation from iPSCs generated from non-T-cells (Fig. 4c). Overall, $1 \times 10^5$ iHPCs generated on average $6.2 \times 10^8$ DL4 cells with 5166-fold increase during T-cell differentiation in 15-cm DL4-coated dishes (Fig. 4d and Supplementary Fig. 6). After the maturation step, DL4 cells from the 4-2 clone completely differentiated into CD4$^-$CD8αβ$^+$CD3$^+$WT1 tetramer$^+$ cells (Fig. 4e). Flow cytometric analysis revealed that matured WT1 tetramer$^+$ cells expressed low levels of CD45RA and CD45RO and high level of CCR7 (Fig. 4f). TCR usages of differentiated T-cells from the 4-2 clone were determined by sequencing for the sequences encoding TCR Vα and Vβ complementary determining region 3 (CDR3) (Fig. 4f). TCR diversity of mature WT1-TCR iCD8αβ T-cells showed nearly monoclonal expression of exogenously introduced Vβ chains. Together, these results demonstrate large-scale production of antigen-specific CD8αβ$^+$tetramer$^+$ T-cells from a TCRαβ-engineered clinical-grade HLA-homozygous iPSC line.

We finally optimized the proliferation culture conditions for TCR-engineered iCD8αβ T-cells. We identified optimal condition for each signal of the TCR complex including TCR activation and cytokine signaling (the 1$^{st}$ and 3$^{rd}$ signals, respectively) (Supplementary Fig. 7a). For 1$^{st}$ signal optimization, a monoclonal antibody against CD3, OKT3, was coated on a well in the presence of retronectin at varying concentrations, and ATP concentrations were examined 3 days after stimulation as a surrogate of cell numbers (Supplementary Fig. 7b). Retronectin has been shown to stimulate peripheral blood T-cell growth in vitro when used together with anti-CD3[28–30]. This screening assay identified the optimal concentrations of OKT3 and retronectin to be 3.0 and 150 μg/ml, respectively (Supplementary Fig. 7c). For 3$^{rd}$ signal optimization, we found that the combination of IL-7, 12, 15, 18, 21, and TL-1A, separately identified in our laboratory, also enhanced the activation of TCR-engineered iCD8αβ T-cells in the Ff condition (Supplementary Fig. 7d). Under these conditions, differentiated CD8αβ$^+$WT1-tetramer$^+$ cells could be activated and proliferated repeatedly without losing the expression of both CD8α and CD8β accompanying on average of 286-fold increase in 14 days after each stimulation (Fig. 5a, b). We also tested if fetal bovine serum (FBS) can be removed from the culture. We screened 17 commercially available serum-free medium and identified ImmunoCult-XF T cell expansion medium and CTS OpTmizer with CTS Immune Cell Serum Replacement could induce

proliferation of TCR-engineered iCD8αβ T-cells 60- and 30-fold over 14 days, respectively (Supplementary Fig. 8a). Flowcytometric analysis demonstrated that those cells proliferated in serum-free medium expressed surface markers similar to conventional, FBS-containing medium (Supplementary Fig. 8b).

Finally, the in vitro and in vivo functions of proliferated iCD8αβ T-cells were evaluated. Flow cytometric analysis showed that the proliferated iCD8αβ T-cells acquired CD45RO expression, but not CCR7 expression (Fig. 5b). The proliferated iCD8αβ T-cells killed target cells loaded with the WT1$_{235–243}$ peptide and a WT-1-expressing cancer cell line NCI-H226 in vitro (Fig. 5c). Flow cytometry for intracellular cytokines demonstrated the production of IFN-γ, TNF, and IL-2 upon the stimulation by iCD8αβ T-cells (Fig. 5d). The WT1-TCR iCD8αβ T-cells proliferated by Ff culture method were also able to control disease progression and showed survival benefit in a xenograft model that had been inoculated with a mesothelioma cell line, NCI-H226 (Fig. 5e, f). Collectively, these results demonstrate a large-scale production of TCR-engineered iCD8αβ T-cells capable of in vitro and in vivo anti-tumor activity without losing the CD8αβ co-receptor expression in clinically relevant Ff conditions.

**Generation and assessment of iCART cells.** In order to assess the functions of iT cells with reference to chimeric antigen receptor (CAR) T-cell therapy, we generated CD19 CAR-expressing iT cells (iCART cells) and evaluated their in vitro and in vivo functions (Fig. 6a). To generate iCART cells, iT cells were retrovirally transduced with a 2$^{nd}$ generation CD19 scFv-4-1BB-CD3ζ, the CAR construct identical to tisagenlecleucel (Fig. 6b). We also transduced iCART cells with a membrane-bound chimeric IL-15[31] (Fig. 6b). The resulting iCART cells were assessed for their surface marker expressions, CD19-specific target cytotoxicity, proliferation, and cytokine productions (Fig. 6c–e, and Supplementary Fig. 9a). iCART cells expressed low levels of CD45RA and CCR7, but high levels of CD45RO and CD62L, similar to parental iT cells (Fig. 6c). iCART cells did not express exhaustion markers, PD1 and TIM3, but expressed LAG3 (Fig. 6c). iCART cells showed CD19-dependent cytotoxicity, where they killed CD19$^+$ NALM-6, but not CD19$^-$ CCRF-CEM cancer cell line (Fig. 6d). In order to evaluate CD19-dependent cell division of iCART cells, we cocultured iCART cells with NALM-6 or CCRF-CEM for 6 days without cytokines and measured their cell division and found that iCART cells divided in the presence of NALM-6, but not in the presence of CCRF-CEM (Fig. 6e). iCART cells also showed CD19-dependent cytokine productions, including IL-2, IFN-γ, TNF, and GM-CSF (Supplementary Fig. 9a). Finally, in vivo anti-tumor effect of iCART cells were evaluated in an established NALM-6 systemic model. iT cells and iCART cells were injected once intravenously into the tumor-bearing mice 4 days after NALM-6 inoculation. To compare the therapeutic efficacy of iCART cells with primary CART cells, we also generated primary T cells expressing CD19-specific CAR (pCART) and injected them into NALM-6-

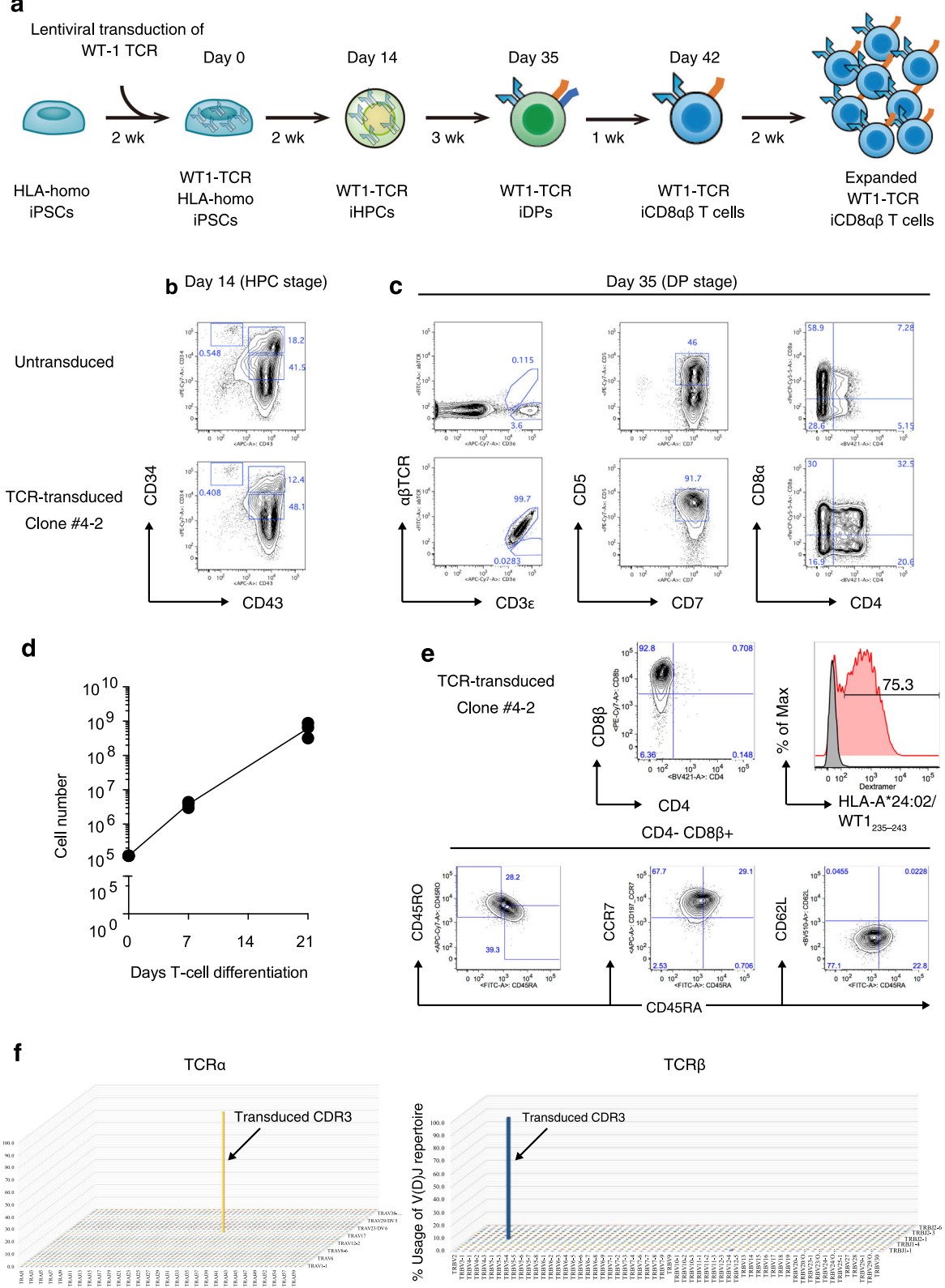

bearing mice (Supplementary Fig. 10a). In PBS control, and iT cell recipients, tumor cells rapidly disseminated throughout the body and all recipients died or were euthanized due to significant weight loss by 3 weeks after cell injections. On the other hand, recipients of iCART cells showed delayed relapse and superior overall survival compared to the control group and recipients of iT cells (Fig. 6f and g). Flowcytometric analysis of bone marrow

cells from iT cell and iCART cell recipients revealed near complete elimination of GFP$^+$CD19$^+$ NALM-6 cells and presence of iCART cells at day 10 and 15 after treatments in iCART cell recipients (Fig. 6h). Notably, two out of five recipients of iCART cells survived relapse-free at least 128 days after treatment. Analysis of those relapse-free recipients showed the presence of iCART cells in the bone marrow, but not in blood cells

**Fig. 4 Generation of CD8αβ⁺ SP T-cells from TCR-transduced HLA-homo iPSCs. a** Scheme of production starting from HLA-homo iPSC to WT1-TCR-transduced iPSC CD8αβ⁺ T-cells. Wk, week; HPCs, hematopoietic progenitor cells; DP, CD4⁺CD8⁺ double-positive; CTLs, CD4⁻CD8αβ⁺ cytotoxic T-cells. **b**, **c** Flow cytometry plot of differentiating untransduced (top) or WT1-TCR transduced (bottom) HLA-homo iPSCs (**b**) 14 days after differentiation (HPC stage) gated on the CD14⁻CD235α⁻ population and (**c**) 35 days after differentiation (DP stage) showing the expression levels of αβTCR and CD3 (left), CD5 and CD7 (middle), and CD8α and CD4 (right) ($n = 3$). **d** Kinetics of total differentiating cell count during T-cell differentiation in a large-scale culture generated from $1 \times 10^5$ WT1-iPSC-HPCs seeded on a DL4/RN-coated 10 cm dish ($n = 3$ independent experiments). Data represent mean ± SD. **e** Flow cytometry plots of mature CD4⁻CD8αβ⁺αβTCR⁺ T-cells generated from WT1-transduced HLA-homo iPSCs at day 42 after differentiation. **f** VDJ frequency of WT1-TCR-transduced iPSC CD8 SP T-cells determined by next-generation sequencing of CDR3 regions of TCRα (left) and TCRβ (right) ($n = 3$ independent experiments).

(Supplementary Fig. 9b). As expected, recipients of pCART cells also showed prolonged survival compared to the control group. Although recipients of iCART cells died of tumor relapse as measured by bioluminescence, some recipients in the pCART cell treatment group began dying with accompanying severe weight loss without evidence of tumor relapse and required euthanasia (Fig. 6f and Supplementary Fig. 10b). These deaths were likely from xenograft-versus-host-disease. These results indicate iCART cells were capable of engrafting, mounting an anti-tumor activity, and persisting after tumor clearance and confer overall survival benefit to tumor-bearing mice comparable to pCART cells (Fig. 6g).

## Discussion

In this study, we report an efficient and scalable method to generate functional CD8αβ⁺ T-cells from iPSCs in clinically relevant Ff culture conditions. This has been accomplished by two approaches: (1) generation of iPSCs from T-cell clones or transduction of TCR to clinical-grade iPSCs derived from non-T-cells for improvement of efficiency and (2) development of Ff differentiation cultures by optimizing each step of differentiation including hematopoietic induction, T-cell commitment, and T-cell activation. We demonstrate successful generation of T-cells from iPSCs under Ff conditions in all stages of differentiation.

Previous studies have shown that HECs or HPCs differentiated from ESCs or iPSCs could be induced to differentiate into the T-cell lineage by co-culturing with OP9 feeder layer-expressing notch ligands, DL1 or DL4, with a limited efficiency of αβ- or γδ-TCR⁺ cell generation[2,3,32]. It has been also demonstrated that generation of iPSC from T-cells (T-iPSCs) and redifferentiation enhance the efficiency of αβTCR-expressing cell generation from iPSCs[4–7,33,34]. Recently, we have also shown that exogenous TCR transduction in clinical-grade iPSCs can be used as a starting material to generate antigen-specific TCR-engineered iCD8αβ⁺ T-cells with in vivo activity, thereby minimizing inter-clonal variations of iPSCs and bypassing laborious iPSC generation processes[9]. We extended these findings by showing that TCR-transduction in iPSCs derived from non-T-cells accelerate T-cell commitment and facilitate αβT-cell generation compared to non-transduced iPSC.

However, all the reports have used different murine feeder cells in several steps during T-cell differentiation from PSCs, making scale-up for clinical application difficult. Most researchers have used OP9 feeder-layers expressing DL1 or DL4 during T-cell differentiation[35]. The uses of xenogenic feeders for clinical application will not be favorable due to safety and scalability issues. In this study, we demonstrate that immobilized-DL4 protein, lymphopoietic cytokines, and supplements induce robust and reproducible T-cell differentiation of iHPCs without OP9 feeder-layer. The identification of SDF1α and CXCR4 signaling as a factor to facilitate T-cell differentiation by augmenting T-cell commitment genes such as BCL11B, the master regulator of T-cell commitment is interesting[36]. These findings are in line with results of previous studies demonstrating a pivotal role of the

CXCR4/SDF1α axis for thymocyte development including β-selection and thymocyte proliferation, highlighting the notion that iPSC-based screening can be generalized to human T-cell biology[37–39]. Although the precise role of SB203580, a selective p38 MAPK inhibitor, in the present culture remains unknown, it may promote survival of iHPCs in the culture as addition of SB203580 alone could improve the survival of differentiating cells from 2nd to 3rd week of culture, where majority of DP cell generation occurs. Moreover, we have shown scale-up of T-cell differentiation culture with a 5166-fold increase. It was seen that $3 \times 10^5$ iPSCs seeded on one well of a 6-well plate generated approximately $6.2 \times 10^8$ DL4 cells after 35 days of differentiation in four 15-cm dishes (Fig. 4d and supplementary Fig. 6).

Previous studies using mature T-cells derived from PSC were typically proliferated on inactivated PBMC feeder cells, which rises safety issues such as virus contamination to allogeneic cell products. The development of the Ff proliferation culture for mature iCD8αβ⁺ T-cells demonstrated in the current study resolves these issues. Importantly, proliferated iCD8αβ T-cells showed in vitro antigen-specific killing of target cells and in vivo anti-tumor activity. In addition, we have demonstrated that generation of iCART cells from iT cells and their therapeutic efficacy in an established systemic tumor model. When compared to tumor-free survival, instead of overall survival, therapeutic efficacy of iCART cells seemed to be inferior to that of pCART cells. One reason for this could be due to the fact that iCART cells did not have CD4⁺ T cells, whereas primary CART cells contained both CD8⁺ and CD4⁺ cells. In accordance with the notion, a recent study has demonstrated that CD4⁺ T cells play a critical role in the anti-tumor effect of CART cell therapy[40]. Efforts to generate CD4⁺ T cells from iPSCs in this platform are underway.

One drawback of the iPSC-T cell-based approach may be the time required to induce iPSC from a T-cell clone and to regenerate T-cells from these clones, which takes approximately 4 months. A potential solution to this issue will be the production of iPSC-T cell banks as "off-the-shelf" T-cell sources. These iPSC-T cell sources would be used as starting materials for downstream modifications such as CAR transduction to construct iCART-cell banks, as demonstrated in Fig. 6. Compared to the manufacturing process of physiologic T-cell immunotherapy, iPSC-T cell bank approach can exclude apheresis procedure from the manufacturing process. The present technology allows us to generate such iPSC-T cell banks. For example, our results have demonstrated generation of over $1 \times 10^9$ iPSC-T cells from $3 \times 10^5$ iPSCs, our minimum culture scale to induce differentiation (Supplementary Fig. 6). The iPSC-T cells can be further expanded approximately 200-fold in the subsequent culture, indicating the feasibility of cell production for multiple patients given that a total of $1 \times 10^9$ cells per patient is required in the current CAR T-cell therapy. To date, we have succeeded to produce $2 \times 10^{10}$ iT cells in ten 1-liter culture devices, a number we believe is sufficient to conduct the initial clinical trial for safety assessment. Development of a large-scale T-cell expansion culture aiming at 100 and more doses is currently underway. In particular,

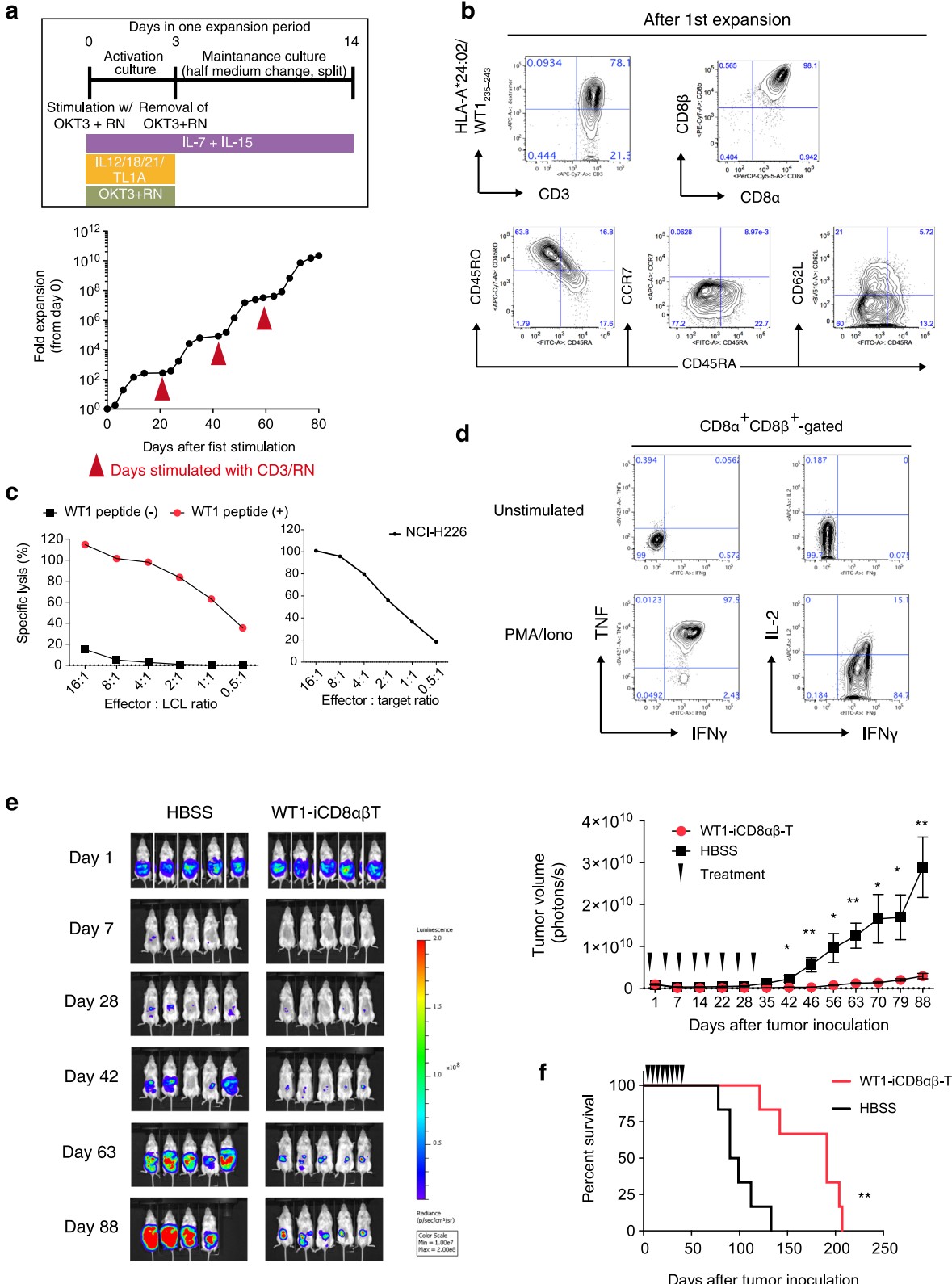

successful commercialization would require development of iT cell proliferation culture in larger apparatus, such as automated stirred-tank bioreactor system. It is also worth mentioning that iPSC-T cells would become an ideal universal cell source for T-cell immunotherapy when other technologies such as HLA-editing for T-cell rejection escape, TCR-truncation for reducing

the risk of GvHD, and gene transductions for NK-cell rejection escape[41–43] are integrated. In summary, the present study demonstrates a highly efficient and scalable platform for generating functional CD8αβ T-cells from iPSCs that can be applied to produce "off-the-shelf" and synthetic T-cell sources for allogeneic T-cell immunotherapy.

**Fig. 5 In vitro and in vivo functions of WT1-TCR-transduced HLA-homo iCD8αβ T-cells. a** Fold proliferation of WT1-TCR iCD8αβ T-cells in an optimized protocol as shown (top) for four sequential rounds of stimulations. Data are representative of two independent experiments. **b** Flow cytometry plots of proliferated WT1-iPSC CD8 SP T-cells representing the expression levels of tetramer and naive/memory T-cell markers ($n = 3$ independent experiments). **c** In vitro cytotoxicity assay of proliferated WT1-TCR iCD8αβ T-cells measured by DELFIA assay using LCL cells loaded with specific peptides as target cells (left) and NCI-H226 (right). Data are representative of two independent experiments. **d** Intracellular cytokine production of WT1-TCR iCD8αβ T-cells after 6 h stimulation with PMA/Ionomycin. Data are representative of two independent experiments. **e, f** In vivo anti-tumor activity of WT1-TCR iCD8αβ T-cells. NSG mice were intraperitoneally inoculated with NCI-H226-expressing luciferase, treated 4 times weekly with HBSS or WT1-TCR iCD8αβ T-cells, and monitored for (**e**) tumor volume and (**f**) survival rate ($n = 5$ animals each). *$P = 0.0016$. Data represent mean ± SEM. *$P < 0.05$; **$P < 0.01$ (Welch's two samples $t$-test (two-tailed) (**e**); log-rank test (two-tailed) (**f**)).

## Methods

**Human induced pluripotent stem cell lines**. PBMC samples were obtained from healthy volunteers who provided written informed consent. The H25-4, H25-31, and GPC3#16-1 iPSC clones were generated from CTL clones by Sendai virus reprogramming vectors as previously described[4,9,33,34]. An HLA homozygous iPSC line Ff-I01s04 was established in CiRA and distributed with informed consent and permission of the Ethical Review Board at CiRA. WT-1 TCR expressing Ff-I01s04 was generated as previously described[9]. Ten WT-1 TCR-transduced iPSC clones were established, expanded, and induced to differentiate into T-cell. At the end of T-cell differentiation, we assessed the expression of the transgenes by flowcytometry and determined the transgene copy number of the clones with robust transduced-TCR expressions. Among them, the clone with the lowest copy number, clone 4-2 was selected for further analysis. All researches using human samples were approved by the Kyoto University School of Medicine ethical committee (no. G590). All iPSCs were maintained in StemFit AK03N (Ajinomoto, Japan) on iMatrix-511 (Matrixome, Japan) as previously described[19].

**Cell lines**. SK-Hep-1 and SK-Hep-1 transduced with GPC3 were maintained in Dulbecco's Modified Eagle Medium (DMEM) supplemented with 10% fetal bovine serum (FBS). HLA-determined B-LCLs (HEV RIKEN Cell Bank) were used as antigen-presenting cells. NCI-H226 cells were purchased from ATCC and maintained in DMEM supplemented with 10% FBS. NALM-6 cells and CCRF-CEM were purchased from RIKEN Cell Bank and ATCC, respectively, and maintained in RPMI1640 medium supplemented with 10% FBS. The cells were cultured at 37 °C in an atmosphere having 5% $CO_2$. The mycoplasma status of the cells were routinely checked and was confirmed to be negative.

**Flow cytometry and antibodies**. All flow cytometry staining were performed in MACS buffer (0.5% BSA and 2 mM EDTA in PBS) for 30 min on ice. For tetramer co-staining, PE- or APC-conjugated HLA-A*02:01–GPC3$_{157-165}$, HLA-A*02:01–HIV$_{26-35}$, HLA-A*24:02-WT-1$_{235-243}$ (Immudex), and HLA-A*24:02 Nef, HLA-A*24:02-Gag tetramers (MBL International) were added to cells at a 1:20 final dilution and stained for 30 min on ice. After washing, cells were stained with additional antibodies for 20 min on ice. PI was added to all samples before analysis. LSRII Fortessa and FACS AriaII or Aria Fusion instrument (BD Biosciences, San Jose, CA) were used for flow cytometry analysis and cell sorting, respectively (FACS Diva 8.0.1). Monoclonal antibodies (specific clones in parentheses) used for surface and intracellular staining of the following molecules were obtained from BioLegend (San Diego, CA): CD3 (UCHT1), CD4 (OKT4), CD5 (UCHT2), CD7 (CD7-6B7), CD8α (SK1), CD14 (4D6), CD27 (O323), CD28 (CD28.2), CD45 (HI30), CD45RA (HI100), CD45RO (UCHL1), CD62L (DREG-56), CCR7 (G043H7), IFN-γ (B27), IL-2 (MQ1-17H12), and TNF-α (Mab11). Antibodies against CD43 (1G10) and CD235α (HIR2) were obtained from BD Biosciences (San Jose, CA). Antibodies against CD34 (4H11) and CD8β (SID18-BEE) were obtained from Abcam (Cambridge, UK) and Thermo Fisher Scientific (Grand Island, NY), respectively. For detection of apoptosis during T-cell differentiation, differentiating cells were stained with Pacific Blue-Annexin V and 7-amino-actinomycin (7-AAD) according to the manufacturer's instructions (BioLegend). Flow cytometry data were processed and analyzed with FlowJo software (Tree Star Inc.).

**Generation of HPCs from iPSCs**. A step-by-step protocol describing the differentiation of iPSCs into T-cells and the proliferation of the regenerated T cells can be obtained at Protocol Exchange[44]. The generation of iCD8αβ T-cells from iPSCs via HPCs was conducted in Ff cultures (Supplementary Fig. 1). To initiate differentiation, iPSCs expanded for 6–7 days on iMatrix-511 in StemFit AK03N were dissociated into single cells using 0.5× TryPLE select (Thermo Fisher Scientific). A total of 3–6 × $10^5$ cells were resuspended in StemFit AK03N supplemented with 10 μM Y-27632 (FujiFilm Wako) and 10 μM CHIR99021 (Tocris Bioscience) and cultured in 6-well ultra-low attachment plates (Corning) for 24 h. After 24 h, the EBs were collected, settled down to the bottom of the tube, and resuspended in 2 ml StemPro-34 (Thermo Fisher Scientific) supplemented with 10 ng/ml penicillin/streptomycin (Sigma), 2 mM Glutamax (Thermo Fisher Scientific), 50 μg/ml ascorbic acid-2-phosphate (Sigma), 4 × $10^{-4}$ M monothioglycerol (MTG, Nacalai), and 1× Insulin-Transferrin-Selenium solution (ITS-G, Thermo Fisher Scientific)

(referred to as EB basal medium), 50 ng/ml recombinant human (rh) BMP-4 (R&D Systems), 50 ng/ml rhVEGF (R&D Systems), and 50 ng/ml bFGF (FujiFilm Wako) per well. After 24 h, 6 μM SB431542 (FujiFilm Wako) was added. After 4 days, the differentiating EBs were collected, washed and resuspended in 2 ml EB basal medium supplemented with 50 ng/ml rhVEGF, 50 ng/ml rhbFGF and 50 ng/ml rhSCF (R&D Systems) per well and cultured for 2 days. After 7 days, the differentiating EBs were again collected, washed, and resuspended in 2 ml EB basal medium supplemented with 50 ng/ml rhVEGF, 50 ng/ml rhbFGF, 50 ng/ml rhSCF, 30 ng/ml rhTPO (PeproTech), and 10 ng/ml FLT3L (PeproTech) per well. From day 7, differentiating cultures were collected and replaced with fresh day 7 medium for every 2–3 days. Cultures were maintained in a 5% $CO_2$/5% $O_2$/90% $N_2$ environment for the first 7 days and in a 5% $CO_2$ environment from day 7 onwards.

**Differentiation of T-cell progenitors**. T-cell differentiation was induced on rhDL4-coated plates prepared one day prior to iHPC seeding. rhDL4/Fc chimera protein solution (10 μg/ml, Sino Biological) was diluted with an equal volume of retronectin (10 μg/ml, TAKARA, Japan), 150 μl of the solution was added to each well of 48-well plates, and incubated overnight at 4 °C. The coating solution was removed just before adding T-cell differentiation medium.

For iHPC seeding, 11–14 EBs were collected and dissociated into single cell by TryPLE Select (Thermo Fisher Scientific) treatment, followed by passing through a 21-G needle (Terumo, Japan) 7 times. A total of 2000 CD235α⁻/CD14⁻/CD34⁺/CD43⁺ cells were FACS-sorted directly into wells of a DL4-coated plate having T-cell differentiation medium composed of αMEM (Thermo Fisher Scientific) supplemented with 15% FBS (Corning), 100× ITS-G (1×), 55 μM 2-Mercaptoethanol (Thermo Fisher Scientific), 50 μg/ml ascorbic acid-2-phosphate, 2 mM Glutamax, 50 ng/ml rhSCF, 50 ng/ml rhTPO, 50 ng/ml rhIL-7, 50 ng/ml FLT3L, 30 nM rhSDF-1α (PeproTech), and 15 μM SB203580 (Tocris Bioscience). A major portion of the medium (80%) was changed every other day. The differentiating cells were transferred to a new DL4-coated plate on day 7 and 1–2 × $10^5$ cells/well were transferred to a new DL4-coated plate on day 14. For large-scale T-cell differentiation cultures, 1 × $10^5$ iHPCs were seeded on a DL4-coated 10 cm dish and the medium was completely changed on days 2 and 4. At day 7, total differentiating cells were harvested and 1 × $10^6$ cells were transferred onto a DL4-coated 15 cm dish. The medium was completely changed on days 9, 11, 14, 16, 18, 19, and 20 without cell transfer at day 14. Cultures were maintained in a 5% $CO_2$ environment.

Serum-free T-cell differentiation was carried out using StemSpan SFEM II (StemCell Technologies), αMEM + BIT medium, and IMDM + BIT medium supplemented with 55 μM 2-Mercaptoethanol (Thermo Fisher Scientific), 50 μg/ml ascorbic acid-2-phosphate, 2 mM Glutamax, 50 ng/ml rhSCF, 50 ng/ml rhTPO, 50 ng/ml rhIL-7, 50 ng/ml FLT3L, 30 nM rhSDF-1α (PeproTech), and 15 μM SB203580 (Tocris Bioscience). αMEM + BIT medium, and IMDM + BIT medium were also supplemented with 100x ITS-G. Serum-free αMEM + BIT medium was composed of αMEM with 20% bovine serum albumin, insulin, and transferrin serum substitute (BIT9500; StemCell Technologies). Serum-free IMDM + BIT medium was composed of IMDM (Thermo Fisher Scientific) supplemented with 20% BIT9500. All other procedures were performed as described above.

**Maturation of T-cell progenitors to CD8 single positive CTLs (iCD8αβ T-cells)**. Day 21 DL4 cells were stimulated with a monoclonal antibody to CD3 (clone: OKT3, eBioscience) at the concentration of 500 ng/ml in maturation medium composed of αMEM, 15% FBS, 100× ITS-G (1×), 50 μg/ml ascorbic acid-2-phosphate, 100× PSG (1×, Sigma), 10 ng/ml rhIL-7, 10 ng/ml rhIL-2 (PeproTech), and 10 nM dexamethasone (Fuji Pharma). The cells were collected, washed, and resuspended in maturation medium without OKT3 after 3 days and incubated for 4 days in an environment containing 5% $CO_2$ at 37 °C.

**Feeder-free proliferation of WT1-TCR iCD8αβ T-cells**. On the day before T-cell activation, 96 or 48-well plates were coated with CD3/retronectin solution composed of 3.0 μg/ml anti-human CD3 MAb (OKT3; eBioscience) and 150 μg/ml retronectin (TAKARA) at 4 °C overnight. A total of 4 × $10^5$ iCD8αβ T-cells/ml were suspended in T-cell activation medium composed of αMEM supplemented with 15% FBS, 100× ITS-G (1×), 50 μg/ml ascorbic acid-2-phosphate, 10 ng/ml rhIL-7, 10 ng/ml rhIL-15 (PeproTech), 20 ng/ml rhIL-21 (PeproTech), 50 ng/ml

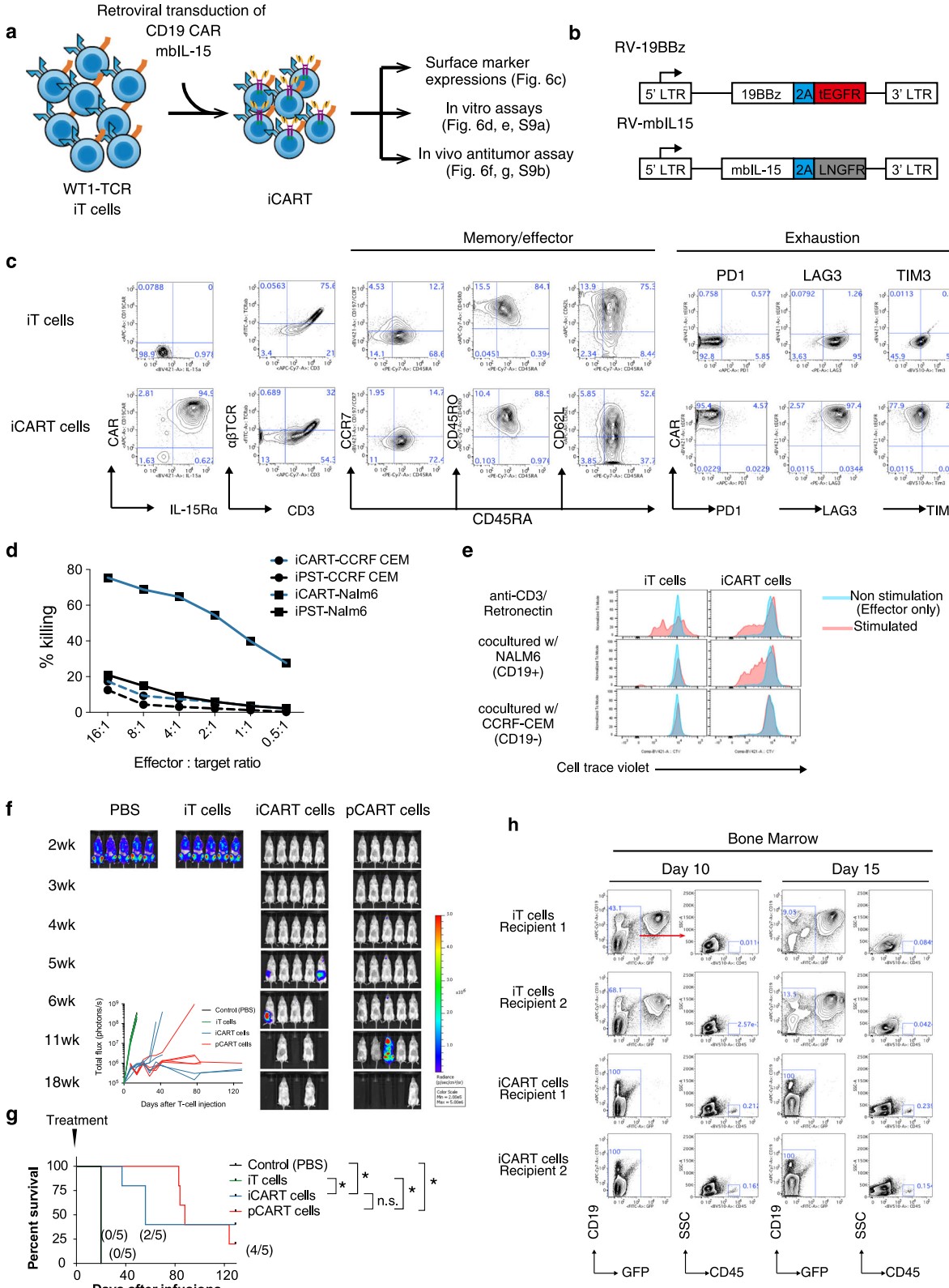

rhIL-12 (Merck), 50 ng/ml rhIL-18 (MBL), 50 ng/ml rhTL-1A (PeproTech), and 10 μM Z-VAD (R&D Systems) and cultured in CD3/RN-coated plates for 3 days. The cultures were collected, washed, and resuspended in proliferation medium composed of αMEM, 15% FBS, 100× ITS-G (1×), 50 μg/ml ascorbic acid-2-phosphate, 10 ng/ml rhIL-7, and 10 ng/ml rhIL-15. Approximately 80% spent medium was changed every 2–3 days, with re-culturing to new wells or larger culture vessels as needed. The cells were counted on days 1, 3, 6, and 14 and re-stimulated as mentioned above for further proliferation after 14 days.

For serum-free expansion, ImmunoCult-XF T Cell Expansion Medium (StemCell Technologies) or CTS OpTmizer T Cell Expansion SFM supplemented with CTS Immune Cell Serum Replacement (Thermo Fisher Scientific) was used as the basal medium and iT cells were activated in the presence of 10 ng/ml rhIL-7, 10 ng/ml rhIL-15 (PeproTech), 20 ng/ml rhIL-21 (PeproTech), 50 ng/ml rhIL-12 (Merck), 50 ng/ml rhIL-18 (MBL), 50 ng/ml rhTL-1A (PeproTech), and 10 μM Z-VAD (R&D Systems) and cultured in CD3/RN-coated plates for 3 days. The resulting cells were collected, washed to remove activation cytokines, and cultured

**Fig. 6 In vitro and in vivo functions of iCART cells. a** A schematic showing generation and assessment of iCART cells from iT cells. **b** Designs of RV-19BBz CAR (top) and RV-mbIL-15 (bottom). This design allows the co-expression of the CAR and truncated EGFR and mbIL-15 and LNGFR from the same LTR promoter by using a self-cleaving P2A sequence. LTR, long terminal repeat, *Porcine teschovirus* self-cleaving 2A sequence. **c** Representative flow cytometry plots of untransduced and 19BBz-transduced iT cells (iCART cells) showing expression levels of CD19 CAR, IL-15Rα, CD3, αβTCR, CD8α, CD8β (left), naïve/memory T-cell markers (middle), and exhaustion markers (right). **d** Cytotoxic activity of iT cells and iCART cells using CD19$^+$ NALM-6 and CD19$^-$ CCRF-CEM as target cells. Data represent two independent experiments. **e** Representative histograms showing cell divisions of CellTrace Violet-labeled iT cells (left) and iCART cells (right) cocultured with CD19 + NALM-6 or CD19- CCRF-CEM (red) or without target cells (blue) for 6 days. Cell divisions of iT cells and iCART cells activated by retronectin/CD3 conditions are shown (top panels). Data represent two independent experiments. **f, g** In vivo anti-tumor activity of iT cells and iCART cells in a systemic tumor model. NOG mice were intravenously inoculated with NALM-6-expressing luciferase 4 days before treatment, treated once intravenously with PBS, $1 \times 10^7$ iT-cells, iCART-cells, or primary CART-cells, and monitored for (**f**) tumor volume and (**g**) survival rate ($n = 5$ mice each). Values in parentheses represent the fraction of mice without tumor relapse. **h** NALM-6-bearing mice were treated with $1 \times 10^7$ iT cells or iCART cells. At 10 and 15 days after treatment, mice were euthanized and bone marrow cells were collected. Presence of iT cell or iCART cells (GFP$^-$CD19$^-$humanCD45$^+$) and NALM-6 (GFP$^+$CD19$^+$) cells were analyzed by flow cytometry. $^*P < 0.05$ (log-rank Mantel–Cox test with Bonferroni corrections, two-tailed).

for an additional 12 days in the same medium supplemented with 10 ng/ml rhIL-7 and 10 ng/ml rhIL-15.

**On-feeder proliferation of iCD8αβ T-cells**. iCD8αβ T-cells were co-cultured with irradiated PBMCs at 1:20 ratio in αMEM supplemented with 15% FBS, 100× ITS-G (1×), 50 μg/ml ascorbic acid-2-phosphate, 2 mM Glutamax, 10 ng/ml rhIL-7, 5 ng/ml rhIL-15, and 2 μg/ml PHA (Sigma). Half the medium was changed every 2–3 days with re-plating as needed.

**TCR repertoire sequencing**. Total RNA was isolated using RNeasy Micro kit (QIAGEN) following the manufacturer's instructions. Bioinformatics analysis was performed using the repertoire analysis software Repertoire Genesis (RG; Ver. 20190912) obtained from Repertoire Genesis Incorporation (Osaka, Japan). RG assigns TRV and TRJ alleles to queries, generates CDR3 sequences, and aggregates their combination patterns. Out-of-frame sequences were excluded from the analyses.

**WT-1 TCR-dependent assays**

*In vitro cytotoxicity assays.* $^{51}$Cr release assays were conducted to evaluate effector cell cytolytic ability. Target tumor cells were loaded with 1.85 MBq $^{51}$Cr for 1 h and 5000 tumor cells were co-incubated with effector cells for 5 h at effector-to-target (E:T) ratios ranging from 20:1 to 2.5:1. Supernatants were harvested and $^{51}$Cr release was quantified using a beta counter (PerkinElmer). Percent lysis was calculated as: % lysis = (experimental lysis−spontaneous lysis)/(maximal lysis −spontaneous lysis) × 100%, where maximal lysis was induced by incubation in a 2% Triton X-100 solution.

*In vivo anti-tumor assays.* Mice used for the study (6-week old female NOD.Cg-Prkdc$^{scid}$Il2rg$^{tm1Sug}$/ShiJic (NOG) mice) were purchased from CIEA (Kanagawa, Japan). The mice were intraperitoneally injected with $5 \times 10^6$ NCI-H226 cells transduced with firefly luciferase. Mice were imaged for tumor bioluminescence regularly by intraperitoneal injection of luciferin. A total of $5 \times 10^6$ iWT1-TCRTs were injected intraperitoneally twice weekly for 4 weeks from 1 h after tumor cell inoculation. HBSS was injected into the control arm. Tumor bioluminescence was detected every 7 days for at least 70 days by IVIS Lumina II (PerkinElmer), after which mice were sacrificed based on body weight reduction criteria (>20% weight loss). Mice were randomly allocated to experimental groups. All animal experiments were conducted under a protocol approved by the institutional Animal Research Committee.

**CD19 CAR-dependent assays**

*Generation of CD19 iCART cells.* The cassette encoding the single-chain antibody targeting CD19 (clone FMC63), CD8 hinge/transmembrane, 4-1BB domain, and the ζ-chain of the CD3 was cloned into the pMYs γ-retroviral backbone[45] (Cell Biolabs, San Diego, CA) to generate the retroviral (RV)-19CAR-BBz vector. pMYs-mbIL-15 plasmid was designed by fusing the full-length IL-15 peptide sequence to the full-length IL-15Rα sequence via a flexible linker as previously described with the IL-2 signal sequence instead of IgE[31]. Both the pMYs-19bbz and the pMYs-mbIL-15 were synthesized by GeneWiz (South Plainfield, NJ).

To generate retroviral vectors, we generated HT1080-based packaging cell lines which produce Moloney murine leukemia virus cores with cat endogenous virus envelop RD114 (FLYRD18) and prepared retroviral supernatants as previously described[46].

For transduction of 19CAR-BBz and mbIL-15, cryopreserved WT-1 TCR iPSC-T cells (iT cells) were recovered for 3 days in the expansion medium composed of IMDM 15% FBS, 100× ITS-G (1×), 50 μg/ml ascorbic acid-2-phosphate, 10 ng/ml rhIL-7, and 10 ng/ml rhIL-15 (PeproTech). iT cells were then activated as described above and cultured for 3 days. At day 4, the activated iT cells were loaded onto

retronectin (TAKARA)-coated 48-well plate (50 μg/ml) that was preloaded with retrovirus supernatant. The 48-well plates were centrifuged for 10 min. at 560 *g* and 37 °C. CD19CAR-transduced iT cells were obtained through MACS separation targeting EGFR. mbIL-15 was retrovirally transduced to the CD19CAR iT cells and purified in the next round of activation/expansion culture to generate iCART cells.

*Generation of primary CART cells.* PBMCs used to prepare primary CART cells were purchased from Precision for Medicine and the use of human samples was approved by ethical committee at Takeda Pharmaceutical Company Ltd. To transduce CAR, T cells were cultured on wells precoated with CD3/retronectin solution composed of 3.0 μg/ml anti-human CD3 MAb (OKT3; eBioscience) and 150 μg/ml retronectin (TAKARA), having T-cell activation medium composed of CTS OpTmizer T Cell Expansion SFM supplemented with CTS Immune Cell Serum Replacement (Thermo Fisher Scientific), supplemented with 400 IU/ml recombinant human IL-2 (PeproTech) for 3 days for activation. The activated T cells were transduced with retroviral vector harboring 19BBz, as described in the above section (Generation of iCART cells). The resulting cells were infused into NALM-6-bearing mice 4 days after transduction.

*In vitro cytotoxicity assays.* For in vitro cytotoxic assays, target cells were labeled with N-SPC Non-radioactive cellular cytotoxicity assay kit (Techno Suzuta) according to the manufacture's instruction. The target cells were pulsed with BM-HT reagent for 37 °C, washed 3 times, and $1 \times 10^4$ cells seeded into a well of a 96-well plate. Effector cells were loaded into the wells at effector-to-target (E:T) ratios ranging from 16:1 to 0.5:1 and co-cultured for 2 h. Then, 20 μl of the co-culture supernatant was mixed with 100 μl of Eu solution and time-resolved fluorescence was measured through the EnVision 2105 multimode plate reader (PerkinElmer). Percent lysis was calculated as: % lysis = (experimental lysis−spontaneous lysis)/ (maximal lysis−spontaneous lysis) × 100%, where maximal lysis was induced by incubation in a detergent solution provided by the manufacturer.

**T-cell division**. To evaluate antigen-specific cell division of iPSC-derived T-cell, we labeled WT1-TCR-transduced iPSC-T cells (iT cells), and CD19 iCART cells with CellTrace Violet (Termo Fisher Scientific). We then stimulated a total of $1 \times 10^5$ iT cells or iCART cells with a total of $1 \times 10^5$ CD19 + NALM-6 or CD19- CCRF-CEM (E:T ratio of 1:1) and measured their CellTrace Violet dilution by the flowcytometer (LSRII Fortessa) after 6 days of culture without activation/expansion cytokines. RN/CD3 activation was used as TCR-dependent cell division.

**Cytokine production**. Culture supernatants as prepared above (T-cell division) were collected after 3, 72, and 144 h of culture to measure the production of human IL-2, interferon-γ (IFN-γ), tumor necrosis factor (TNF), GM-CSF, and Granzyme B using BD Cytometric Bead Array Flex Set System (BD Bioscience).

**In vivo anti-tumor assays**. Mice used for the study (7-week old female NOD.Cg-Prkdc$^{scid}$Il2rg$^{tm1Sug}$/ShiJic (NOG) mice) were purchased from CIEA (Kanagawa, Japan). The mice were intravenously injected with $5 \times 10^5$ NALM-6 cells transduced with firefly luciferase. Mice were imaged for tumor bioluminescence regularly by intraperitoneal injection of luciferin. A total of $1 \times 10^7$ iWT1-TCRTs, iCART, pCART cells were injected once intravenously 4 days after tumor cell inoculation. PBS was injected into the control arm. Tumor bioluminescence was detected every 7 days for at least 6 weeks by IVIS Lumina II (PerkinElmer), after which mice were sacrificed based on body weight reduction criteria (>20% weight loss). Mice were randomly allocated to experimental groups. All animal experiments were conducted under a protocol approved by the institutional Animal Research Committee.

**Quantitative real-time PCR.** Sorted CD235a⁻/CD14⁻/CD34⁺/CD43⁺ cells were cultured on DL4-coated plates at $1 \times 10^5$ cells/well in a 6-well plate with differentiation medium and collected after 48 and 96 h followed by a PBS wash. Cells were lysed and total RNA was isolated using RNeasy Micro kit according to the manufacturer's instruction. cDNA was prepared using Veso cDNA synthesis kit (Thermo Fischer Scientific). Real-time quantitative PCR was performed on a QuantStudio 7 Flex (Applied Biosystems). PCR amplification was carried out in triplicate using pre-designed gene-specific Taqman Gene Expression Assay probes (Supplementary Table 1) and 2× Taqman Fast Advanced Master Mix (Applied Biosystems). Relative expression levels were calculated as ΔΔCt relative to *ACTB*.

**Statistical analysis.** In all figure legends, *n* represents the number of independent experiments conducted and data are represented as mean ± standard deviation (SD) or mean ± standard error of the mean (SEM) as indicated. Statistical analysis was performed using GraphPad Prism software and *P*-values were calculated by ordinary one-way ANOVA with Tukey's multiple comparison test. For Fig. 5e, f, tumor volume and survival analysis were tested by Welch's two samples *t*-test and log-rank test, respectively. For Fig. 6g, log-rank test was performed to calculate *P*-values. *$P < 0.05$; **$P < 0.01$; and ***$P < 0.001$ were considered statistically significant.

**Reporting summary.** Further information on research design is available in the Nature Research Reporting Summary linked to this article.

## Data availability
The authors declare that all data that support the findings of this study are available in this article and are provided as a Source File Data file or from the corresponding author upon reasonable request. Source data are provided with this paper.

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

## Acknowledgements

We thank Prof. Shinya Yamanaka (Kyoto University) for providing the HLA homozygous iPSC line and giving critical advice for our research work. We also thank Prof. Hiromitsu Nakauchi (The University of Tokyo) and Prof. Naoko Takasu (Kyoto University) for providing the iPSC lines; Dr. Seigo Izumo (T-CiRA) for giving critical advice; Dr. Wang Bo (Kyoto University); Dr. Keiko Koga, Dr. Masashi Yamada, and Dr. Sujatha Mohan (T-CiRA); Mr. Shuichi Kitayama, Mr. Kohei Ohara, and Mr. Akito Tanaka (Kyoto University); Ms. Ayako Kumagai, Sanae Kamibayashi, Ms. Eri Imai, and Ms. Katsura Noda (Kyoto University); Ms. Mariko Sekiguchi, Ms. Maki Numazaki, Ms. Yuka Maruyama, Ms. Yuki Watanabe, Ms. Ai Kikuchi, Ms. Xuewei Song, Ms. Hitomi Takakubo, Ms. Megumi Tada, and Ms. Mizuki Kobayashi (T-CiRA) for technical and administrative assistances. The entire study was conducted in accordance with the Declaration of Helsinki and permitted by the institutional ethical board of Kyoto University. This work was supported in part by the Ministry of Education, Culture, Sports, Science and Technology of Japan (23591413, 15H04655, 15J05263, 26293357, and 18K16085), Japan Agency for Medical Research and Development (Project for Development of Innovative Research on Cancer Therapeutics, and Core Center for iPS Cell Research), the Takeda-CiRA collaboration program, and the collaborative research grant of Thyas Co., Ltd.

## Author contributions

S.I., Y.Y., Y.K., and S.K. designed the study; S.I., Y.Y., Y.K., and S.K. interpreted the data; S.I., Y.Y., Y.K., S.A., M.K., T.Sato., T.U., N.Y., Y.M., Y.B., M.Y., and Y.Kassai performed the experiments; S.I., S.A., M.K., A.H., and S.K. analyzed the data; Y. Mishima, A.M., T.S., K.N., M.T., T.N., and M.Y. provided critical materials, protocols or advice for the experiments; S.K. supervised the study; and S.I. and S.K. wrote the manuscript.

## Competing interests

The authors declare the following financial competing interests. S.K. is a founder, shareholder, and chief scientific officer at Thyas Co., Ltd. and received research fundings from Takeda Pharmaceutical Co., Ltd., Kyowa Hakko Kirin Co., Ltd., Sumitomo Chemical Co., Ltd., and Thyas Co., Ltd. S.A., M.K., T. Sato., Y.B., T.S., K.N., M.T., Y. Kassai, and A.H. are employees of Takeda Pharmaceutical Co. Ltd. Y.Y. is an employee of Thyas Co. Ltd. The remaining authors declare no competing financial interests. These authors and all other authors declare no other competing interests.
