## [Peer Review File · Nature Communications]

Reviewer #1 (Remarks to the Author):

The manuscript by Iriguchi et al. describes a set of culture optimizations for the differentiation T-iPSC (T cell-derived induced pluripotent stem cells) or T cell receptor (TCR)-transduced iPSCs, into T cells in vitro. The work is well done and the results support the conclusion that human T-iPSCs can be differentiated into functional T cells using plate-bound Dll4-Fc plus retronectin along with a cocktail of hemato/lymphopoietic cytokines, starting from embryoid body- differentiated T-iPSCs that have reached the CD34+ CD43+ stage. None of these steps or conditions are novel, and in essence represent a technical refinement of the known Dll4-dependent induction of T cell development. Nevertheless, the demonstration that human iPSCs can be fully differentiated into T cells using a stromal cell-free system is notable, however, the use of fetal bovine serum (FBS) severely detracts from the main claim that the optimized protocol represents a fully clinically ready approach. The authors re-discovered the importance of SDF1a (CXCL12), which is well known as a key chemokine signal supporting early T cell development, and the fact that blocking p38 can improve in vitro T cell development, which is also well known and previously established, in the OP9-DL1 system at the preTCR DN-DP-SP differentiation stages. The remaining novelty of the work comes from the demonstration that a clinically relevant HLA homozygous iPSC line can be used for the generation of T cells.

Reviewer #2 (Remarks to the Author):

In the submitted manuscript, Iriguchi et al. report on the preparation of TCR-transgenic T-cells from induced pluripotent stem cells (iPSC). The manuscript and presented data are of interest for the readership of Nature Communications however, there are multiple conceptual and practical concerns that need to be addressed in order to enhance clarity for the reader and to substantiate the scientific and translational merit of the paper.

Major comments:

1. Anti-tumor function of iPSC-derived CD8 α β + T-cells in vitro:

- The authors only present functional data against target cells that have been pulsed with peptide and it is unclear whether the iPSC-derived T-cells are able to recognize target cells (including tumor cell lines or primary tumor cells) that express the cognate antigen. These data need to be included.
- T-cell effector functions like proliferation (e.g. by CFSE assay) and cytokine analysis (ELISA/Luminex) are missing. The stimulation of cytokine production by PMA/ionomycin is typically used as a positive control and is not informative as to whether antigen recognition is indeed capable of activating the T-cells and to induce cytokine secretion. These data need to be included.
- It is unclear how the effector function of iPSC-derived T-cells compares to the parental T-cell clone from which the transgenic TCR has been derived (and to 'conventional' T cells that recognize the same antigen). This comparison ought to be included in order to enhance clarity for the reader and to demonstrate that the iPSC strategy does not lead to a loss of anti-tumor function.
- There is a lack of information on the phenotype of the iPSC-derived T-cells including expression level of the transgenic TCR (and endogenous TCR?), a basic flow panel including CD45RA, CD45RO, CD62L, as well as activation/exhaustion markers, costimulatory ligands, adhesion molecules, etc. These data ought to be included in order to allow the reader to assess how closely these induced T-cells resemble physiologic T-cells.

2. Anti-tumor function of iPSC-derived CD8 α β + T-cells in vivo:

The presented in vivo data are insufficient and do not reflect the state of the art in the field.

- A major problem is that the tumor model (and anti-tumor response) is essentially confined to a particular anatomical compartment (tumor cells and T-cells are administered i.p.) and does not reflect a systemic tumor. Rather, a model with systemic tumor and systemic administration of the T-cells (e.g. through i.v. tail vein injection) should be presented.
- The in vivo experiments lack a control treatment group that receive iPSC-derived T-cells that do not express the transgenic CAR as a reference. This control is essential and needs to be included.
- The in vivo engraftment and persistence of the transferred T cells seems to be very limited. Despite the multiple doses of T cells that are administered, their anti-tumor activity is very limited and rather disappointing. Typically, a single dose of tumor-reactive T cells, administered i.v., is capable of exerting a significant anti-tumor effect against a systemic tumor.
- Data demonstrating that the transferred T-cells actually persist after transfer, for example through analysis in peripheral blood, bone marrow and tumor lesions, are lacking.

3. Genetic engineering strategy, cell culture protocol and yield:

- It is unclear to this reviewer if the iPSC-derived T-cells still contain the endogenous TCR (as the iPSCs were induced from a T cell clone) – please clarify.
- The time required to induce pluripotent stem cells from a T-cell clone and to (re)generate T-cells from these induced pluripotent stem cells, is in excess of 6 weeks and the overall yield from one induced pluripotent stem cell is approximately 70.000 fold. In current clinical trials of adoptive T-cell therapy, T-cell doses between 1×10^6 /kg bodyweight and 1×10^7 /kg bodyweight of the patient are administered requiring a total T-cell yield of approximately 1×10^8 and 1×10^9 per patient (assuming 10kg body weight). Can the presented process be scaled up sufficiently to provide enough T cells for 1 or multiple patients?
- With conventional T-cells, manufacturing takes approx. 7-9 days with lenti- or gamma-retroviral gene transfer (and this process can further be shortened to 1-2 days through the use of virus-free gene transfer strategies). The authors ought to put their proposed strategy into perspective in comparison to these 'conventional approaches'. In particular, the authors ought to comment on whether these iPSCs constitute indeed a perpetual source of T-cells, i.e. how long can these iPSCs be cultured and what is the maximum anticipated yield per iPSC-cell over its life span?
- The authors ought to comment as to whether indeed iPSCs constitute a universal and off the shelf T-cell source in the absence of genetic modification of the endogenous T-cell receptor locus (risk of GvHD) and HLA-molecules (rapid immune rejection by the host patient).

4. Selection of an iPSC-cell clone for T-cell preparation:

In the submitted manuscript, the authors selected clone 4-2 to derive CD8 $\alpha\beta$ ⁺ T-cells. It is unclear which criteria were applied to select this clone and whether similar results could be obtained when a different iPSC-clone would have been selected. Also, data ought to be included addressing as to whether the iPSC-derived T-cells resemble the phenotype and anti-tumor reactivity of the original T-cell clone the T-cell receptor has been derived from. These data are lacking but are required to enhance clarity for the reader and to substantiate the findings.

5. The manuscript is well written and the main and supplemental figures are visually appealing however, at multiple times the figure call outs in the text do not match the composition of panels in the actual figures, and several figures are not called out at all.

Reviewer #3 (Remarks to the Author):

Iriguchi, Yasui et al describe an improved method of generating functional T-cells from T-iPSC bypassing the need of feeder cells and utilizing coated DLL4 and a cocktail of soluble factors to help improve expansion of committed T-cell precursors. Overall, the study is performed at a very high technical level and will be of interest to a wide audience. Clarifying the following points would help strengthen the story and make it more appealing.

1. Figure 2 demonstrates addition of SDF1a and a p38 inhibitor SB203580 is critical for proper expansion of committed thymocytes from iPSC. The authors show addition of this combination increases relative expression of T-lineage factors but it is unclear whether this change in gene expression simply reflects enrichment for DP T-cells shown in Fig 2b. Does the SS combination reduce apoptosis of DP cells or increases their proliferation Or does it promote the DN->DP transition somehow? Do either DN or DP cells (or both) express SDF1 receptor CXCR4? What is the contribution of each individual component (SDF1a alone vs SB alone) and is there a synergy between the two? Clarifying the mechanism of SS-mediated enhancement of T-cell differentiation is critical to this study
2. What is the role of retronectin in DLL4 and OKT3 coating? Retronectin is traditionally used to facilitate gammaretroviral transduction of activated T-cells but its contribution in the described protocol is unclear. Does it promote T-cell adhesion to DLL4 or OKT3-coated surface or also induces T-cell differentiation? This point has to be clarified either experimentally or by referencing existing literature.
3. The authors indicate CD8 as co-stimulatory receptor providing Signal 2, which is unconventional. Classic model indicates CD8 contributes to Signal 1 by bridging MHC recognition with Lck/Fyn-mediated ITAM phosphorylation of the zeta chain, the main trigger of Signal 1. Conventional Signal 2 receptors include IgSF and TNFRSF receptors CD28/ICOS and CD27/4-1BB/OX40 etc. Supplementary Fig 6 shows optimization of T-cell expansion by titrating OKT3 + retronectin, which are supposed to produce only Signal 1, which is known to induce T-cell energy, especially in low concentrations of OKT3. This mode of activation is used in Figure 5. Would the authors see an improvement when mixing (lower concentrations of) OKT3 with anti-CD28 or anti-4-1BB antibodies with retronectin? Comparison with CD3/CD28 beads would not be the most appropriate due to variation in several factors, such as Ab concentration, presence of retronectin, spherical shape, etc.

Point-by-point response to reviewers

The authors thank the reviewers for their critical reading of our manuscript and insightful comments. We were hopeful that all reviewers regarded this work appropriate for the readership of *Nature Communications*. We have revised the manuscript thoroughly to reflect on the reviewer's comments and suggestions. Additional data presented in the revised manuscript have substantiated our main claim. Each comment as shown in black and our response as shown in red are listed below:

Reviewer #1 (Remarks to the Author):

The manuscript by Iriguchi et al. describes a set of culture optimizations for the differentiation T-iPSC (T cell-derived induced pluripotent stem cells) or T cell receptor (TCR)-transduced iPSCs, into T cells in vitro. The work is well done and the results support the conclusion that human T-iPSCs can be differentiated into functional T cells using plate-bound Dll4-Fc plus retronectin along with a cocktail of hemato/lymphopoietic cytokines, starting from embryoid body-differentiated T-iPSCs that have reached the CD34⁺ CD43⁺ stage. None of these steps or conditions are novel, and in essence represent a technical refinement of the known Dll4-dependent induction of T cell development. Nevertheless, the demonstration that human iPSCs can be fully differentiated into T cells using a stromal cell-free system is notable, however, the use of fetal bovine serum (FBS) severely detracts from the main claim that the optimized protocol represents a fully clinically ready approach. The authors re-discovered the importance of SDF1a (CXCL12), which is well known as a key chemokine signal supporting early T cell development, and the fact that blocking p38 can improve in vitro T cell development, which is also well known and previously established, in the OP9-DL1 system at the preTCR DN-DP-SP differentiation stages. The remaining novelty of the work comes from the demonstration that a clinically relevant HLA homozygous iPSC line can be used for the generation of T cells.

To address the reviewer's comment, we performed additional experiments to test the requirement of FBS in T-cell differentiation and T-cell expansion cultures by either replacing FBS with bovine serum albumin (BSA) or commercially available serum-free medium as shown in Supplementary Figure 3 and 8. We found that FBS could be replaced with BSA supplemented with insulin and transferrin (BIT) for T-cell differentiation. Moreover, the choice of basal medium appeared to be critical as the use of IMDM supplemented with BIT and StemSpan medium, which is also composed of IMDM, reduced the frequency of DP cells after differentiation. For T-cell expansion culture, we found that, among 17 commercially available serum-free medium, ImmunoCult-XF and OpTmizer T-cell expansion medium were capable to replace FBS while maintaining the surface marker expressions similar to cells expanded in the FBS-containing medium.

Reviewer #2 (Remarks to the Author):

In the submitted manuscript, Iriguchi et al. report on the preparation of TCR-transgenic T-cells from induced pluripotent stem cells (iPSC). The manuscript and presented data are of interest for the readership of Nature Communications however, there are multiple conceptual and practical concerns that need to be addressed in order to enhance clarity for the reader and to substantiate the scientific and translational merit of the paper.

Major comments:

1. Anti-tumor function of iPSC-derived CD8 $\alpha\beta$ + T-cells in vitro:

The authors only present functional data against target cells that have been pulsed with peptide and it is unclear whether the iPSC-derived T-cells are able to recognize target cells (including tumor cell lines or primary tumor cells) that express the cognate antigen. These data need to be included.

To address the reviewer's comment, we performed an additional experiment where WT-1-TCR transduced iPSC-T cells (iT cells) were cocultured with NCI-H226 and assessed for their target killing. As shown in Figure 5c, iT cells killed target cells in an increasing effector-to-target ratio manner, indicating iT cells are capable of recognizing the cognate antigen.

T-cell effector functions like proliferation (e.g. by CFSE assay) and cytokine analysis (ELISA/Luminex) are missing. The stimulation of cytokine production by PMA/ionomycin is typically used as a positive control and is not informative as to whether antigen recognition is indeed capable of activating the T-cells and to induce cytokine secretion. These data need to be included.

To address the reviewer's comment, we performed additional experiments where iT cells or iT cells transduced with CD19 CAR (iCART cells) were coculture with CD19+ NALM-6 cells or CD19-CCRF-CEM cells and assessed for their cell trace violet dilution and cytokine productions as shown in Figure 6 and supplementary figure 9a. We observed CD19-specific cell division and cytokine productions of iCART cells.

It is unclear how the effector function of iPSC-derived T-cells compares to the parental T-cell clone from which the transgenic TCR has been derived (and to 'conventional' T cells that recognize the same antigen). This comparison ought to be included in order to enhance clarity for the reader and to demonstrate that the iPSC strategy does not lead to a loss of anti-tumor function.

We agree with the reviewer's comment and have attempted to perform additional experiments. However, we no longer have parental T-cell clones as they were used in other studies (Minagawa et

al., Cell Stem Cell, PMID: 30449714 and Kawai et al., in revision). These works together with other publications have demonstrated that iPSC technology did not lead to a loss of TCR-dependent target recognitions.

There is a lack of information on the phenotype of the iPSC-derived T-cells including expression level of the transgenic TCR (and endogenous TCR?), a basic flow panel including CD45RA, CD45RO, CD62L, as well as activation/exhaustion markers, costimulatory ligands, adhesion molecules, etc. These data ought to be included in order to allow the reader to assess how closely these induced T-cells resemble physiologic T-cells.

To address the reviewer's comment, we performed an additional experiment where we assessed expressions of CCR7, CD45RA, CD45RO, CD62L, PD1, LAG3, TIM3 on WT-1 TCR-transduced iPSC-T cells and iCART cells by flow cytometry as shown in Figure 6c. This experiment show that iPSC-T cells express a pattern of cell surface markers similar to effector memory T cells. They did not express PD1 and TIM3, but expressed LAG3. These findings indicate that expression profile of iPSC-T cells is similar to, but not identical to physiologic T-cells.

2. Anti-tumor function of iPSC-derived CD8 $\alpha\beta$ + T-cells in vivo:

The presented in vivo data are insufficient and do not reflect the state of the art in the field.

A major problem is that the tumor model (and anti-tumor response) is essentially confined to a particular anatomical compartment (tumor cells and T-cell systemic administration of the T-cells (e.g. through i.v. tail vein injection) should be presented.

To address the reviewer's comment, we performed a series of additional experiments where CD19 CAR was transduced to iPSC-T cells and injected them once intravenously into a systemic tumor model (NALM-6 cells injected i.v. four days before treatment) to evaluate in vivo antitumor activity through assessments of tumor cell imaging, survival, and persistence (new Figure 6). NALM-6 rapidly disseminated in mice treated with PBS, 5 million iT cells and 10 million iT cells from 2 weeks after treatment. Conversely, treatment with iCART cells demonstrated superior antitumor activity, including reduced tumor burden, significant increase in survival, and longer persistence. Two out of five mice treated with 10 million iCART cells showed no tumor relapse and survived at least 128 days after treatment. These findings add an important conclusion that iPSC-T cells have antitumor activity similar to physiologic CAR T-cells.

The in vivo experiments lack a control treatment group that receive iPSC-derived T-cells that do not express the transgenic CAR as a reference. This control is essential and needs to be included.

To address the reviewer's comment, we performed additional experiments where iPSC-T cells that do not express CD19 CAR were also injected into the NALM-6 model side by side with iCART cells as described above (Figure 6 f-h).

The in vivo engraftment and persistence of the transferred T cells seems to be very limited. Despite the multiple doses of T cells that are administered, their anti-tumor activity is very limited and rather disappointing. Typically, a single dose of tumor-reactive T cells, administered i.v., is capable of exerting a significant anti-tumor effect against a systemic tumor.

To address the reviewer's comment, we performed an additional experiment where iCART cells were injected once i.v. into a well-established NALM-6 model to examine the potential of iPSC-T cells. We have demonstrated that iCART cells injected once i.v. were capable of eliminating NALM-6 and exerting a significant antitumor activity in the model.

Data demonstrating that the transferred T-cells actually persist after transfer, for example through analysis in peripheral blood, bone marrow and tumor lesions, are lacking.

To address the reviewer's comment, we performed an additional experiment where bone marrow cells of NALM-6 mice 10 and 15 days after injection of 10 million iT cells or iCART cells were analyzed by flowcytometry for detection of human CD45+ cells. We found that iCART cells persisted for at least 15 days after injection in the bone marrow. In addition, the end point analysis of relapse-free mice at day 128 after treatment showed that iCART cells remained in the bone marrow, indicating they persisted in these mice.

3. Genetic engineering strategy, cell culture protocol and yield:

It is unclear to this reviewer if the iPSC-derived T-cells still contain the endogenous TCR (as the iPSCs were induced from a T cell clone) – please clarify.

The reviewer raises an important point. As T-iPSCs are derived from a T-cell clone, iPSC-T cells contain the rearranged endogenous TCR with additional rearrangements of TCR α chain because of the fact that differentiating cells gain the expression of recombinase-activating-gene (RAG). In order to circumvent this issue, we have recently demonstrated that CRISPR out RAG2 inhibited these additional rearrangements and the loss of specificity (Minagawa et al., Cell Stem Cell, 2018, PMID: 30449714).

The time required to induce pluripotent stem cells from a T-cell clone and to (re)generate T-cells

from these induced pluripotent stem cells, is in excess of 6 weeks and the overall yield from one induced pluripotent stem cell is approximately 70,000 fold. In current clinical trials of adoptive T-cell therapy, T-cell doses between 1×10^6 /kg bodyweight and 1×10^7 /kg bodyweight of the patient are administered requiring a total T-cell yield of approximately 1×10^8 and 1×10^9 per patient (assuming 10kg body weight). Can the presented process be scaled up sufficiently to provide enough T cells for 1 or multiple patients?

The reviewer makes a good point. In this study, we found that addition of both SDF1 α and SB203580 allowed us to expect approximately 3,000-fold expansion during T-cell differentiation as shown in Figures 2 and 4. Using this protocol, for example we are now able to generate more than 1×10^9 iPSC-T cells starting from 3×10^5 iPSCs. In addition, we have also developed an iPSC-T-cell expansion culture that allows us to expect on average 200-fold expansion after each TCR activation. These findings support the notion that the present process has been sufficiently scaled-up to generate enough number of iPSC-T cells for multiple patients.

With conventional T-cells, manufacturing takes approx. 7-9 days with lenti- or gamma-retroviral gene transfer (and this process can further be shortened to 1-2 days through the use of virus-free gene transfer strategies). The authors ought to put their proposed strategy into perspective in comparison to these 'conventional approaches'. In particular, the authors ought to comment on whether these iPSCs constitute indeed a perpetual source of T-cells, i.e. how long can these iPSCs be cultured and what is the maximum anticipated yield per iPSC-cell over its life span?

The authors agree with the reviewer's comment and included these considerations into the context. We added additional text to the discussion to propose our strategy. We will not intend to culture iPSCs for a long period of time to produce iPSC-T cells. For clinical trials, we are going to generate a large number of iPSC-T cells from iPSC cell bank at low passage number to construct an iPSC-T cell bank. We are not going to continue iPSC cultures after the differentiation begins. At the beginning of each iPSC-T cell campaign production, a new tube from the iPSC cell bank will be recovered and used for differentiation. The present method will be a foundation to produce such iPSC-T cell banks.

The authors ought to comment as to whether indeed iPSCs constitute a universal and off the shelf T-cell source in the absence of genetic modification of the endogenous T-cell receptor locus (risk of GvHD) and HLA-molecules (rapid immune rejection by the host patient).

We put these considerations into the discussion of the text. We believe that genetic modification of

TCR locus and HLA-molecules as well as molecules to reduce NK-cell-mediated rejections are mandatory to constitute a universal and “off-the-shelf” iPSC T cells

4. Selection of an iPSC-cell clone for T-cell preparation:

In the submitted manuscript, the authors selected clone 4-2 to derive CD8 $\alpha\beta$ + T-cells. It is unclear which criteria were applied to select this clone and whether similar results could be obtained when a different iPSC-clone would have been selected. Also, data ought to be included addressing as to whether the iPSC-derived T-cells resemble the phenotype and anti-tumor reactivity of the original T-cell clone the T-cell receptor has been derived from. These data are lacking but are required to enhance clarity for the reader and to substantiate the findings.

We apologize for the lack of information with regard to the clone selection. We added text in the Method to substantiate the criteria as shown in red font. To address the second part of the comment, we have attempted to perform additional experiments where antitumor activity of iPSC-T cells is compared to those of the original T-cell clones. However, since we no longer have the original T-cell clones as they were used in the other studies, such experiments are impossible to conduct. Instead, in this study we assessed in vitro functions of iPSC-T cells in the exactly same system as the original clones were assessed and found their functions were comparable to the original clones. The results are shown in Figure 3. Phenotype of iPSC-T cells are shown in Supplementary Figure 5.

5. The manuscript is well written and the main and supplemental figures are visually appealing however, at multiple times the figure call outs in the text do not match the composition of panels in the actual figures, and several figures are not called out at all.

We apologize for the confusions. We revised the manuscript carefully to resolve the issue.

Reviewer #3 (Remarks to the Author):

Iriguchi, Yasui et al describe an improved method of generating functional T-cells from T-iPSC bypassing the need of feeder cells and utilizing coated DLL4 and a cocktail of soluble factors to help improve expansion of committed T-cell precursors. Overall, the study is performed at a very high technical level and will be of interest to a wide audience. Clarifying the following points would help strengthen the story and make it more appealing.

1. Figure 2 demonstrates addition of SDF1 α and a p38 inhibitor SB203580 is critical for proper expansion of committed thymocytes from iPSC. The authors show addition of this combination increases relative expression of T-lineage factors but it is unclear whether this change in gene expression simply reflects enrichment for DP T-cells shown in Fig 2b. Does the SS combination

reduce apoptosis of DP cells or increases their proliferation Or does it promote the DN->DP transition somehow? Do either DN or DP cells (or both) express SDF1 receptor CXCR4? What is the contribution of each individual component (SDF1a alone vs SB alone) and is there a synergy between the two? Clarifying the mechanism of SS-mediated enhancement of T-cell differentiation is critical to this study

To address the reviewer's comment, we performed an additional experiment where transitions of cell surface marker expressions (CD4, CD5, CD7, and CD8) and apoptosis markers (Annexin V and 7AAD) were assessed during T-cell differentiation by flow cytometry in the presence or absence of SDF1 and/or SB203580. We found that SDF1 and SB203580 in synergy to improve not only cell yield of differentiating cells, but also frequency of DP cells as shown in Figure 2d. Detection of apoptosis markers indicated addition of both reagents and SB203580 alone could improve the live cell frequency as determined by Annexin V-/7-AAD- from day 14 to day 21, where the majority of DP cells emerge. These findings suggest that the SS combination also has a role in reducing apoptosis of DP cells.

2. What is the role of retronectin in DLL4 and OKT3 coating? Retronectin is traditionally used to facilitate gammaretroviral transduction of activated T-cells but its contribution in the described protocol is unclear. Does it promote T-cell adhesion to DLL4 or OKT3-coated surface or also induces T-cell differentiation? This point has to be clarified either experimentally or by referencing existing literature.

The reviewer raised an important point. As the reviewer mentioned, retronectin has been used to facilitate gamma-retroviral transduction of T-cells. In the T-cell differentiation, retronectin has been shown to improve in vitro T-cell differentiation of cord-blood hematopoietic stem and progenitor cells (PMID: 25157026). With regard to the role of retronectin, a recombinant fragment of fibronectin, on T-cell activation, previous studies have demonstrated that the very late activation Ag (VLA) 4 on T cells mediate T cell adhesions as well as modulate costimulation of TCR/CD3-complex thereby promote in vitro activation of T cells through a fragment of its receptor retronectin (PMID: 7673711 18464805 and 24497917). We have put these references in the revised text in red font.

3. The authors indicate CD8 as co-stimulatory receptor providing Signal 2, which is unconventional. Classic model indicates CD8 contributes to Signal 1 by bridging MHC recognition with Lck/Fyn-mediated ITAM phosphorylation of the zeta chain, the main trigger of Signal 1. Conventional Signal 2 receptors include IgSF and TNFRSF receptors CD28/ICOS and

CD27/4-1BB/OX40 etc. Supplementary Fig 6 shows optimization of T-cell expansion by titrating OKT3 + retronectin, which are supposed to produce only Signal 1, which is known to induce T-cell energy, especially in low concentrations of OKT3. This mode of activation is used in Figure 5. Would the authors see an improvement when mixing (lower concentrations of) OKT3 with anti-CD28 or anti-4-1BB antibodies with retronectin? Comparison with CD3/CD28 beads would not be the most appropriate due to variation in several factors, such as Ab concentration, presence of retronectin, spherical shape, etc.

We apologize for the misstatement that CD8 provides the Signal 2 of T-cell activation. We have corrected the figure accordingly in the revised manuscript. We have also tested if addition of Signal 2 receptor such as CD28 and others into RN/CD3 coating could improve the outcome and found that some of them improved antitumor activity and persistence in vivo (manuscript in preparation).

Reviewer #1 (Remarks to the Author):

The Authors have directly addressed all the technical concerns regarding their ability to differentiate T cells using a defined serum-free and scalable system. The work is of high rigour and clearly establishes a plate-bound system for the generation of T-lineage cells from PSCs.

Reviewer #2 (Remarks to the Author):

Comments for the authors:

The authors have made a comprehensive and credible effort to address my critique from the first round of review. I am content with the revisions that have been implemented into the manuscript, however, there are several major issues that remain and that ought to be addressed prior to publication.

Major comments:

1. Comment first round of review:

'It is unclear how the effector function of iPSC-derived T-cells compares to the parental T-cell clone from which the transgenic TCR has been derived (and to 'conventional' T cells that recognize the same antigen). This comparison ought to be included in order to enhance clarity for the reader and to demonstrate that the iPSC strategy does not lead to a loss of anti-tumor function.'

Comment second round of review:

The author's response is not satisfactory. This is a critical point, as it is important for readers to assess what the relative anti-tumor function is of iPSC-derived WT1T cells compared to the parental T-cell clones. Indeed, protocols to amplify and perpetuate such T-cell clones are known and established in the field (Riddell et al. Journal of Immunological Methods 1990) and it is unfortunate that apparently, these T-cell clones have been spent in other projects while this manuscript has been under revision. This is still a major issue and one effort that the authors could undertake is to transfer this T-cell receptor into primary human T-cells and compare their anti-tumor function to the iPSC-derived T-cells that express the same TCR.

2. Comment first round of review:

'Anti-tumor function of iPSC-derived CD8 $\alpha\beta$ + T-cells in vivo: The presented in vivo data are insufficient and do not reflect the state of the art in the field. A major problem is that the tumor model (and anti-tumor response) is essentially confined to a particular anatomical compartment (tumor cells and T-cell systemic administration of the T-cells (e.g. through i.v. tail vein injection) should be presented.'

Comment second round of review:

This experiment and comparison is not acceptable. The new data with CD19 CAR T are misleading and there is no point in taking T-cells that have been derived from the iPSC-system and then modify them with a CAR construct. Rather the authors ought to take a CAR T-cell, revert it into an iPSC cell and then derive novel T-cell progeny that carries the same CAR construct. Accordingly, this set of experiments ought to be rather removed in order to maintain clarity for the reader.

3. Comment second round of review:

Retention of the endogenous T-cell receptor in TCR modified TiPSCs:

Thank you for the clarification, it is important to also explicitly state this fact in the manuscript text to maintain clarity for the reader.

4. Comment first round of review:

'The time required to induce pluripotent stem cells from a T-cell clone and to (re)generate T-cells

from these induced pluripotent stem cells, is in excess of 6 weeks and the overall yield from one induced pluripotent stem cell is approximately 70.000 fold. In current clinical trials of adoptive T-cell therapy, T-cell doses between 1×10^6 /kg bodyweight and 1×10^7 /kg bodyweight of the patient are administered requiring a total T-cell yield of approximately 1×10^8 and 1×10^9 per patient (assuming 10kg body weight). Can the presented process be scaled up sufficiently to provide enough T cells for 1 or multiple patients? ‘

Comment second round of review:

This again is an important point that needs to be put into perspective for the readership of the article. If indeed 1×10^9 iPSC T-cells can be derived from 3×10^5 iPSCs, then this equals the number of T-cells that is typically required to treat one (a single patient) with a T-cell product. The authors ought to clarify to what extent they can scale the production of the iPSCs and what effort is actually necessary to treat larger cohorts of patients (100 or 1000 patients) as claimed in the manuscript. This could be done for example through a table that is included as part of the result section or in the discussion.

Reviewer #3 (Remarks to the Author):

My comments were addressed appropriately.

Point-by-point response to reviewers

We thank the reviewer for constructive inputs to our manuscript. Each comment as shown in black and our response as shown in red are listed below. We made corresponding changes in the revised manuscript by yellow highlight.

Reviewer #1 (Remarks to the Author):

The Authors have directly addressed all the technical concerns regarding their ability to differentiate T cells using a defined serum-free and scalable system. The work is of high rigour and clearly establishes a plate-bound system for the generation of T-lineage cells from PSCs.

We appreciate the reviewer's positive comments on the revised manuscript.

Reviewer #2 (Remarks to the Author):

Comments for the authors:

The authors have made a comprehensive and credible effort to address my critique from the first round of review. I am content with the revisions that have been implemented into the manuscript, however, there are several major issues that remain and that ought to be addressed prior to publication.

Major comments:

1. Comment first round of review:

'It is unclear how the effector function of iPSC-derived T-cells compares to the parental T-cell clone from which the transgenic TCR has been derived (and to 'conventional' T cells that recognize the same antigen). This comparison ought to be included in order to enhance clarity for the reader and to demonstrate that the iPSC strategy does not lead to a loss of anti-tumor function.'

Comment second round of review:

The author's response is not satisfactory. This is a critical point, as it is important for readers to assess what the relative anti-tumor function is of iPSC-derived WT T cells compared to the parental T-cell clones. Indeed, protocols to amplify and perpetuate such T-cell clones are known and established in the field (Riddell et al. Journal of Immunological Methods 1990) and it is unfortunate that apparently, these T-cell clones have been spent in other projects while this manuscript has been under revision. This is still a major issue and one effort that the authors could undertake is to transfer this T-cell receptor into primary human T-cells and compare their anti-tumor function to the iPSC-derived T-cells that express the same TCR.

We completely agree with the concern raised by the Reviewer. We are aware that loss of TCR-specificity as a consequence of T-iPSC generation and differentiation is an important concern,

but we believe that this would be outside the scope of this manuscript because we have previously shown comparisons in therapeutic efficacy between iPSC-T cells and primary T cells transduced with WT-1 TCR (Minagawa *et al.*, *Cell Stem Cell*, 2018, PMID: 30449714). Please refer to Figures 3 and 4 in the article. Another group has also addressed this issue in an article and found no evidence of TCR-specificity loss by iPSC generation and differentiation starting from a WT-1 T-cell clone (Maeda *et al.*, *Cancer Research*, 2016, PMID: 27872100). To extend these findings, our group has investigated this concern in a separate manuscript using T-iPSCs derived from a CTL clone specific to a HIV epitope, which is now under revision in a journal (Kawai *et al.*, in revision).

We believe that the efficacy of iPSC-T cells could be compared with that of primary T cells if both cells express the same CD19-specific 19BBz CAR, the construct identical to tisagenlecleucel. In attempt to address this question, we have conducted an *in vivo* experiment where we infused primary human T cells expressing CD19 CAR or iPSC-T expressing CD19 CAR into NALM-6-bearing NSG mice and compared their survival for therapeutic efficacy. As you can see in Figure 6 g and f, efficacy of iPSC-CART cells, as measured by tumor-free survival, appeared to be less than those of primary CART cells. We speculate that presence of CD4⁺ T cells in the primary CART cells may be a reason for better efficacy as previously reported (Sommermeyer *et al.*, *Leukemia*, 2016, PMID:26369987). Generation of CD4⁺ T cells has not been achieved in our system and is a next step for our group. This point is included in the discussion section of the manuscript.

2. Comment first round of review:

‘Anti-tumor function of iPSC-derived CD8 α β ⁺ T-cells *in vivo*: The presented *in vivo* data are insufficient and do not reflect the state of the art in the field. A major problem is that the tumor model (and anti-tumor response) is essentially confined to a particular anatomical compartment (tumor cells and T-cell systemic administration of the T-cells (e.g. through i.v. tail vein injection) should be presented.’

Comment second round of review:

This experiment and comparison is not acceptable. The new data with CD19 CAR T are misleading and there is no point in taking T-cells that have been derived from the iPSC-system and then modify them with a CAR construct. Rather the authors ought to take a CAR T-cell, revert it into an iPSC cell and then derive novel T-cell progeny that carries the same CAR construct. Accordingly, this set of experiments ought to be rather removed in order to maintain clarity for the reader.

We appreciate the critical comment from the Reviewer. We would like to emphasize that the purpose of this experiment was to test efficacy of iPSC-T cells in a systemic tumor model, a type of model you kindly suggested in the first round of review. We chose 19BBz CAR as a preclinical model

because the NALM-6-bearing mice model is an accepted systemic tumor model to test the efficacy of experimental CART cells in the field (Eyquen *et al.*, *Nature*, 2017, PMID: 28225754). The reason why we did not choose WT1 TCR model is that there are no systemic tumor models reported for HLA24:02-restricted WT-1 TCR.

3. Comment second round of review:

Retention of the endogenous T-cell receptor in TCR modified TiPSCs:

Thank you for the clarification, it is important to also explicitly state this fact in the manuscript text to maintain clarity for the reader.

We have added this point on page16 of the manuscript

4. Comment first round of review:

‘The time required to induce pluripotent stem cells from a T-cell clone and to (re)generate T-cells from these induced pluripotent stem cells, is in excess of 6 weeks and the overall yield from one induced pluripotent stem cell is approximately 70.000 fold. In current clinical trials of adoptive T-cell therapy, T-cell doses between 1×10^6 /kg bodyweight and 1×10^7 /kg bodyweight of the patient are administered requiring a total T-cell yield of approximately 1×10^8 and 1×10^9 per patient (assuming 10kg body weight). Can the presented process be scaled up sufficiently to provide enough T cells for 1 or multiple patients?’

‘Comment second round of review:

This again is an important point that needs to be put into perspective for the readership of the article. If indeed 1×10^9 iPSC T-cells can be derived from 3×10^5 iPSCs, then this equals the number of T-cells that is typically required to treat one (a single patient) with a T-cell product. The authors ought to clarify to what extent they can scale the production of the iPSCs and what effort is actually necessary to treat larger cohorts of patients (100 or 1000 patients) as claimed in the manuscript. This could be done for example through a table that is included as part of the result section or in the discussion.

We are grateful to the reviewer for the comment. To date, we have had experiences to produce up to 2×10^{10} iT cells in this system in 10 larger culture devices (1 L each) by one operator and we are capable of producing more cells if we increase the number of operators. We assume that this extent of scale up will be sufficient to conduct the initial clinical trial with around 20 patients aiming to evaluate safety and efficacy, if possible. We are aware that this platform is not suitable to treat large patient cohorts (1000 patients) and therefore development of automated bioreactor systems would become critical. We added these points in the discussion section.

Reviewer #3 (Remarks to the Author):

My comments were addressed appropriately.

We appreciate the reviewer's positive comment on the revised manuscript.